# Beyond Markovian Drifts: Action-Biased Geometric Walks with Memory for Personalized Summarization

**Parthiv Chatterjee**
KDM Lab
Dhirubhai Ambani University
Gandhinagar, Gujarat, India
`202421011@dau.ac.in`

**Asish Joel Batha**
KDM Lab
Dhirubhai Ambani University
Gandhinagar, Gujarat, India

**Tashvi Patel**
Adani University
Ahmedabad, Gujarat, India

**Sourish Dasgupta**
KDM Lab
Dhirubhai Ambani University
Gandhinagar, Gujarat, India
`sourish_dasgupta@dau.ac.in`

**Tanmoy Chakraborty**
IIT Delhi, India
IIT Delhi Abu Dhabi, UAE
`tanchak@iitd.ac.in`

## Abstract

Personalized document summarization helps readers focus on the "content-of-interest", a *subjective* and *time-variant* quantity. Recent news recommendation and summarization models often assume that preferences follow a *memoryless or short-memory random walk* on interaction graphs, i.e., a Markovian diffusion seeded at the latest interaction or compressed into a short hidden state or prompt. We ask whether such a hypothesis also holds for personalized summarization. To this end, we propose `Walk2Pers`, a lightweight encoder-decoder framework that extends the walk view with *action-conditioned geometric steps*, decomposed into a (i) a *magnitude* controlling shift strength, and (ii) an *orientation* capturing continuity vs. novelty. The process is mediated by dual memory lanes that reinforce consistent interests while suppressing disinterest, and is augmented with a drift term for summary requests. We show theoretically that such structured walks approximate first-order action-conditioned kernels, and empirically validate the hypothesis on three benchmark datasets – PENS, OpenAI-Reddit, and PersonalSum. Using PerSEval, a personalization metric with strong human correlation, `Walk2Pers` outperforms specialized personalized summarizers by an average of $0.41 \uparrow$, and strong LLM baselines (DeepSeek-R1-14B, LLaMA-2-13B, Mistral-7B, Zephyr-7B) by $0.22 \uparrow$. Our analyses further confirm cross-domain robustness ($0.19 \uparrow$ over the best LLM) and stability on long histories. Together, these results support viewing personalized summarization as an *action-biased geometric walk with memory*, offering both interpretability and efficiency.

## 1 Introduction

With the problem of information overload, personalized summarization has become essential for tailoring updates to a reader's individual interests, especially in multi-aspect documents covering diverse topics (Dasgupta et al., 2024). Existing approaches typically rely on *static* persona attributes (Dou et al., 2021; He et al., 2022; Li et al., 2023). Yet, datasets capturing user reading behaviors, such as MS/CAS PENS (Ao et al., 2021), reveal that user preferences evolve over time and shift across fine-grained subtopics. This creates difficulties even for state-of-the-art (SOTA) Large Language Models (LLMs), which show degraded performance when long user-interaction histories are embedded in prompts for in-context personalized summarization (Liu et al., 2024; Patel et al., 2024). This suggests that both existing personalized summarizers and LLMs struggle to capture subtle, action-specific user interactions within user logs.

A natural follow-up question is *how do user preferences predictably evolve over time?* In most recommendation and personalized summarization systems, user preference evolution has been modeled using simple Markovian assumptions, the most basic case being *pure Random Walk*, where each new preference state is treated as an isotropic perturbation of the previous one, without any action semantics conditioning. A stronger variant is the *Action-conditioned Random Walk*, which biases each step by the most recent action type, while still remaining memoryless. More recently, graph-based *Random Walk with Restart* (RWR) methods, including Personalized PageRank and its extensions, have been widely applied in news recommendation, where user interests are modeled as a diffusion process seeded at the last interaction. Notably, S-Walk (Qiu et al., 2022) restructures the transition kernel to improve session-level modeling, and D-RDW (Zhang et al., 2025) diversifies restart paths to mitigate popularity bias. Although these models remain competitive as *lightweight graph baselines*, extensive benchmarking under the same evaluation regimes (e.g., MIND, Adressa) has shown them to be outperformed by neural encoders such as NAML (Wu et al., 2019a), NRMS (Wu et al., 2019b), and EBNR (Okura et al., 2017), which aggregate user click histories via CNNs, Transformers, or GRUs. However, these models reduce long histories into compressed embeddings, providing only shallow memory. Similarly, preference-prompted mid-sized LLM summarizers can condition on past interactions, although their windowed prompts impose a hard memory cap, with no persistent reinforcement or suppression. Collectively, graph-based diffusion, short-memory neural encoders, and prompt-based LLMs fall under the *Markovian Drift Hypothesis (MDH) – each new state depends primarily on the most recent interaction, overriding long-horizon action dynamics*.

In this paper, we test how well MDH holds for personalized summarization. As an alternative extension to MDH, we propose the Structured Walk Hypothesis (SWH). SWH decomposes the preference state update due to each user interaction (*click*, *skip*, *summarize*) into (i) a *magnitude*, controlling the strength of the action-specific nudge, and (ii) an *orientation*, determining whether the evolution follows the existing trajectory (continuity) or departs into a new direction (novelty). In this way, SWH is an *action-conditioned geometric walk*. To go beyond MDH, SWH incorporates dual memory lanes that reinforce consistent interests while suppressing disinterest (for a comparative table of SOTA MDH models see Table 10). We propose `Walk2Pers`, a personalized summarization model, as a concrete realization of SWH. To test the adequacy of MDH, we pose three research questions. **RQ1:** *Is MDH sufficient for modeling preference evolution in summarization?* **RQ2:** *Do dual memory lanes and action-conditioned geometric steps with magnitude–orientation decomposition, provide systematic and complementary gains over MDH variants?* **RQ3:** *How does `Walk2Pers`, as an instantiation of SWH, compare against short-memory neural encoder augmented personalized summarizers, prompt-personalized LLMs, and oracle summarizers?*

For **RQ1** and **RQ2**, we benchmark MDH-based neural encoder models, along with our own simple action-conditioned Short-Memory Drift (SMD) and an action-category sensitive Action-Gated drift (AGD), on the next user behavior prediction task. We show that these MDH baselines do not sustain asymmetric reinforcement / suppression or disentangle continuity from novelty. In contrast, structured walks with dual memory and geometric decomposition achieve systematic gains, with `Walk2Pers` outperforming the best AGD model by **0.07/0.12/0.18**↑ w.r.t AUC/MRR/nDCG@5/10 metrics. For **RQ3**, we assess downstream personalized summarization. On the PENS dataset, `Walk2Pers` surpasses specialized summarizers (PENS-NAML, NRMS, EBNR, GTP, SP) by an average of **0.42/0.36/0.43**↑ across personalization metrics (PSE-JSD/SU4/METEOR). It also outperforms four mid-sized LLMs (Zephyr-7B, LLaMA2-13B, Mistral-7B, DeepSeek-R1-14B) under 2-shot+history prompting (the best configuration), with DeepSeek-R1 lagging by **0.20/0.29/0.35**. Together, these results demonstrate that while MDH models can be competitive in recommendation-style ranking, SWH with explicit magnitude-orientation decomposition, dual memories, and drift, faithfully captures evolving user preferences for personalized summarization.

## 2 RELATED WORK

**Personalized Summarization Evaluation.** Personalized summarization has been increasingly recognized as essential for tailoring updates to a reader's interests, especially when documents cover multiple aspects. Existing evaluations of summarization quality (e.g., ROUGE, METEOR, BLEU) do not explicitly account for personalization. Recent work (Vansh et al., 2023; Dasgupta et al., 2024) has emphasized the need for personalization-aware metrics. EGISES proposed a framework

for evaluating semantic shift under personalization, while Dasgupta et al. (2024) introduced **PerSE-val**, which we adopt in this paper as it correlates strongly with human judgment.

**Datasets for personalized summarization.** To study evolving user preferences, we require datasets with (i) temporal orders of user interactions, (ii) user-specific expected summaries for shared contents, and (iii) diverse, shifting topics and subtopics. In this direction, the MS/CAS **PENS** dataset (Ao et al., 2021) contains click/skip logs with multi-aspect articles and user-target summaries (per trajectory: 13.6 topics; 52.83 sub-topics, with topic change rate of 0.77). It has become a benchmark for testing personalization-aware models (Ao et al., 2021; Song et al., 2023; Lian et al., 2025). **PersonalSum** (Zhang et al., 2024), The Norwegian dataset derived from Adressa, augments news interaction logs with personalized gold summaries. It highlights preference drift and is suitable for multilingual evaluation. **OpenAI-Reddit** (Völske et al., 2017) provides long-range user interaction traces (posts and comments) with subjective summaries. This non-news multi-domain dataset stresses long-horizon dependencies and temporal drift, and is used for cross-domain generalizability test. Dataset details are in Appendix C.

**Personalized Summarization Models.** Most existing personalized summarizers rely on *static* user personas, as in GSUM, CTRLSum, TMWIN, and Tri-Agent (Dou et al., 2021; He et al., 2022; Kirstein et al., 2024; Xiao et al., 2024). Dynamic extensions such as PENS (Ao et al., 2021) incorporate external news-recommendation encoders like NRMS Wu et al. (2019b), NAML Wu et al. (2019a), and EBNR Okura et al. (2017), while GTP (Song et al., 2023) leverages latent editing controls and SCAPE (Lian et al., 2025) blends content with stylistic features. However, these approaches remain within the scope of the MDH. The few-shot LLM personalization (Patel et al., 2024) achieves competitive performance against such models, but is ultimately stalled by prompt length and memory constraints. In contrast, our work explores SWH, where preference evolution is modeled as a memory-aware, action-conditioned geometric walk. `Walk2Pers` serves as one concrete instantiation capturing evolving user histories beyond the limits of MDH-style summarizers.

# 3 METHODOLOGY: USER PREFERENCE EVOLUTION REPRESENTATION

## 3.1 PREFERENCE DATA AS USER–INTERACTION GRAPH (UIG)

We represent user histories as a **User–Interaction Graph** (UIG), a directed acyclic graph $G = \langle N, E \rangle$ where the node set $N$ consists of three disjoint types: (i) **u-nodes** $u^{(t_0)}$ denoting a user at initial timestep $t_0$, (ii) **d-nodes** $d^{(t_i)}$ representing documents interacted at timestep $t_i$, and (iii) **s-nodes** $s_u^{(t_j)}$ representing user-specific summaries requested or generated at time $t_j$ for a document viewed at $t_{j-1}$. The edge set $E$ encodes user actions: $a_d^{(t_i)} \in \{\texttt{click}, \texttt{skip}, \texttt{summarize}\}$ on documents, and $a_s^{(t_j)}$ as the follow-up $\texttt{summGen}$ action connecting a document $d^{(t_{j-1})}$ to its summary $s_u^{(t_j)}$. A user trajectory $\tau_u$ is then a time-ordered sequence of such interactions, beginning at $u^{(t_0)}$. Each trajectory can be decomposed into *behavior duplets* $b_u^{(t_i)} = \langle a^{(t_i)}, tl^{(t_i)} \rangle$, pairing an action with its tail node $tl^{(t_i)}$. The UIG $\mathcal{T}$ is a pool of trajectories, used as $\mathcal{T}_{\text{train}}$ for training and $\mathcal{T}_{\text{test}}$ for evaluation. For UIG construction, see Appendix C.4; Figure 4.

While UIG captures rich temporal detail, directly modeling its raw structure quickly becomes computationally expensive and noisy over long horizons. This is particularly challenging in personalized summarization, where fine-grained shifts in preference must be retained without overwhelming the model. Recent work in sequential recommendation suggests that *hierarchical abstractions* improve long-horizon accuracy by condensing low-level interactions into higher-order behavioral units (Cho & Hyun, 2023; Ou et al., 2025; Zhu et al., 2023; Pan & Wang, 2021; Zhang et al., 2020). Motivated by this, we adopt a bi-level hierarchy – the **u-layer** records raw interactions: user nodes $u^{(t_0)}$, document nodes $d^{(t)}$, and summary nodes $s^{(t)}$, connected by action edges $a^{(t)}$, and the **b-layer** abstracts these into *behavior duplets* $b_{u_j}^{(t)} = \langle a^{(t)}, tl^{(t)} \rangle$, represented as **b-nodes**. Sequential dependencies are captured by *nextBehavior* edges. This abstraction provides a compact yet expressive substrate for modeling preference evolution. Having established the representational basis, we now ask a fundamental question: *how do preferences evolve along b-layer trajectories?*

Table 1: **SWH (vs. MDH)**: (i) explicit **trajectory modeling** across $b$-nodes, (ii) dual **memory lanes** $(h^+, h^-)$ for reinforcement and suppression, and (iii) **action-aware updates** per $a^{(t)}$.

| Aspect | Markovian Drift Hypothesis | Structured Walk Hypothesis |
|---|---|---|
| History usage | Collapsed into last state or seed $q$ | Explicit trajectory across b-nodes |
| Memory | None or short-lived (hidden states, attention) | Dual memory lanes $(h^+, h^-)$ for reinforcement/suppression |
| Action conditioning | Minimal (recency, weak prompt) | Explicit action-aware updates per $a^{(t)}$ |
| Step dynamics | *Magnitude:* implicit (via GRU/attention weights/prompt tokens) | *Magnitude:* $\mathrm{mag}(a^{(t)})$ |
| | *Orientation:* not modeled explicitly | *Orientation:* $\theta(a^{(t)})$ |
| Long-term preferences | Forgotten beyond 1 step | Persist through memory-conditioned updates |
| Interpretability | Opaque embeddings/hidden states | Geometric (continuity vs. novelty) and stochastic (controlled walk) |

## 3.2 MODELING USER PREFERENCE EVOLUTION

We model preference evolution on the b-layer. Each visited b-node $b_u^{(t)}$ is associated with a latent preference embedding $e_{b,u}^{(t)} \in \mathbb{R}^d$ summarizing user $u$'s state after timestep $t$. We first state the prevailing *Markovian Drift* assumption, then contrast it with our *Structured Walk Hypothesis*.

### 3.2.1 MARKOVIAN DRIFT HYPOTHESIS (MDH)

Under MDH, the next state depends only on the immediately preceding state (or a short compressed representation), while longer histories are discounted or collapsed into a recency prior $q$:

$$e_{b,u}^{(t+1)} \ = \ f\big(e_{b,u}^{(t)}, a^{(t)}, q\big) + \epsilon^{(t)}, \qquad \epsilon^{(t)} \sim \mathcal{N}(0, \Sigma(a^{(t)})). \tag{1}$$

Here $\Sigma(a^{(t)}) \in \mathbb{R}^{d \times d}$ controls how stochastic drift spreads across embedding dimensions. The action-conditioning lets skips be noisier than focused clicks. A pure Random Walk (PRW) is: $f(e_{b,u}^{(t)}, a^{(t)}, q) = e_{b,u}^{(t)}$. Short-memory neural encoders (NAML, NRMS, EBNR) and prompt-personalized LLMs also fit this umbrella by compressing history into short-term aggregates.

### 3.2.2 STRUCTURED WALK HYPOTHESIS (SWH)

Evidence from PENS (Ao et al., 2021), PersonalSum (Zhang et al., 2024), and OpenAI-Reddit (Völske et al., 2017) indicates long-horizon dependencies: clicks reinforce, skips suppress, and summary requests induce systematic drifts. We posit the **SWH**: *preference evolution is a structured, action-conditioned geometric walk with memory*:

$$e_{b,u}^{(t+1)} = e_{b,u}^{(t)} + \Phi\Big(a^{(t)}, \text{trajectory-context}\Big) + \Psi\big(h_t^+, h_t^-\big) + \Delta\Big(a^{(t)}\Big), \tag{2}$$

where $\Phi(\cdot)$ decomposes trajectory-context into a *momentum* direction $u^{(t)}$ (continuity) and an *orthogonal novelty* direction $o^{(t)}$, $\Psi(\cdot)$ aggregates asymmetric reinforcement/suppression via $(h_t^+, h_t^-)$, and $\Delta(\cdot)$ captures special action drifts (e.g., summGen). This decomposition is inspired by advances in trajectory-based dynamic embeddings (e.g., JODIE (Kumar et al., 2019a)) and angle-based relational models (e.g., RotatE, ChronoR (Sun et al., 2019; Anshelevich et al., 2021)), but adapted to the summarization setting with explicit action bias and memory. To illustrate, frequent climate-policy clicks keep movement near $u^{(t)}$ and accumulate in $h_t^+$; repeated skips of celebrity content load $h_t^-$ and downweight entertainment; issuing summGen after dense reports triggers $\Delta(\cdot)$, nudging toward concise representations. This view generalizes random walks into a controlled diffusion governed by state, action, and memory traces (Kumar et al., 2019b; Balcer & Lipinski, 2025).

**SWH Model Family.** Refining Eq. equation 2, we obtain a generic *structured walk* family:

$$e_{b,u}^{(t+1)} = e_{b,u}^{(t)} + \underbrace{\mathrm{mag}(a^{(t)})\big(\cos\theta(a^{(t)})\, u^{(t)} + \sin\theta(a^{(t)})\, o^{(t)}\big)}_{\Phi:\ \text{geometric step: continuity vs. novelty}}$$
$$+ \underbrace{\Psi(h_t^+, h_t^-)}_{\Psi:\ \text{dual memory: reinforcement vs. suppression}} + \underbrace{\delta \cdot \mathbb{I}[a^{(t)} = \texttt{summGen}]}_{\Delta:\ \text{summary-specific drift}}. \tag{3}$$

Here $\mathrm{mag}(a^{(t)})$ scales the step (single click = small shift; repeated clicks = larger shift). The angle $\theta(a^{(t)})$ steers between $u^{(t)}$ and $o^{(t)}$: small $\theta$ (so $\cos\theta \approx 1$, $\sin\theta \approx 0$) favors continuity; larger $\theta$ increases novelty. $\Psi(h_t^+, h_t^-)$ persists asymmetric signals from past actions, and the drift $\delta$ (active for

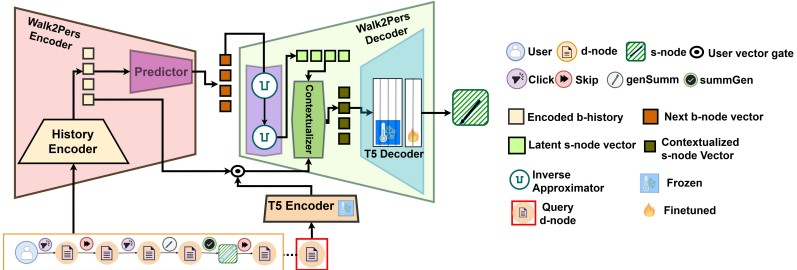

Figure 1: **Walk2Pers:** Novel instantiation of SWH – b-node embedding is formed from the u-layer and fed into the **SWH-Encoder**; **Predictor** estimates the next b-node, the embedding of which is fed into the **Inverse Approximator**, which extracts the latent summary (s-node); **Contextualizer** computes cross-attention of latent s-node, user history, with query document; **T5-decoder** finetuned (top-layers only) to generate personalized summary. b-node embedding details in Figure 2.

summGen) captures shifts due to specific interest signal. Different models may parameterize mag, $\theta$, and $\Psi$ differently, but all share this decomposition. In the next section, we present Walk2Pers as one concrete instantiation of this family. Theoretical relationship to the MDH is in Appendix D.

## 3.3 WALK2PERS MODEL AS SWH INSTANTIATION

### 3.3.1 WALK2PERS ENCODER: HISTORY ENCODING WITH STRUCTURED WALKS

We instantiate SWH (Eq. 3) in Walk2Pers by encoding user trajectories as action-aware b-cells, augmented with dual memories for persistence and geometric steps for continuity–novelty tradeoff.

**Action-biased B-cell composition.** Each d-node and s-node is initialized with a T5-base encoder (Raffel et al., 2020), while each action at timestep $t_i$ is represented as a 4-d one-hot vector (*click*, *skip*, *summarize*, *summGen*). A b-cell fuses the action with its tail-node content: $c_{\text{tl}}^{(t_i)} = \tanh\big(f^{(a,t_i)}\big) \odot e_{\text{tl}}^{(t_i)}; e_{b,u}^{(t_i)} = \tanh\big(W_b \cdot c_{\text{tl}}^{(t_i)}\big)$. where $f^{(a,t)}$ is an *action-conditioned gate*. We borrow the $\text{AGD}(\cdot)$ function of the baseline Action-Gated Drift model (AGD) (Section 4.2; Appendix E.1), which generates action-specific $f^{(a,t)}$ for action type (click, skip, summarize, summGen). Here, click strengthens the fused representation, skip weakens it, and summGen anchors it to a summary node.

**Dual memories and drift (Realization of $\Psi$ and $\Delta$).** To capture long-term asymmetries, Walk2Pers models history as a linear combination of dual memory lanes: $\mathbf{h}^{(t)} = \omega^{(t)}\mathbf{h}^{(+,t)} + (1-\omega^{(t)})\mathbf{h}^{(-,t)}$ ($\omega$ is learnable scalar), where $\mathbf{h}^{(+)}$ accumulates reinforcement signals (clicks), and $\mathbf{h}^{(-)}$ accumulates suppression signals (skips) as follows:

$$\mathbf{h}^{(+,t_i)} = \mathbf{h}^{(+,t_{i-1})} + m^{(t_i)} \odot \mathbf{c}_{\text{tl}}^{(t_i)}; \ \mathbf{h}^{(-,t_i)} = \mathbf{h}^{(-,t_{i-1})} \odot (1 - m^{(t_i)}) + \mathbf{c}_{\text{tl}}^{(t_i)}. \tag{4}$$

In case where the action is summGen, triggering a drift vector $\Delta^{(t)}$, $\mathbf{h}^{(+,t_i)} = \mathbf{h}^{(+,t_{i-1})} + m^{(t_i)} \odot \Delta^{(t_i)}$, where $\Delta^{(t)} = (\mathbf{I} - e_{\text{tl}}^{(t-1)}) \cdot e_{\text{tl}}^{(t)}$, and $m^{(t_i)} = \textbf{SoftMax}\Big(\mathbf{W}_h \cdot \mathbf{h}^{(t_{i-1})} + \mathbf{W}_c \cdot \mathbf{c}_{tl}^{(t_i)}\Big)$. $\Delta$ nudges the preference state toward condensed representations since summarize−summGen is a stronger positive signal. The corresponding action-gate is then applied as: $f^{(a,t_i)} = \text{AGD}(\mathbf{e}_a, t_i) \odot \mathbf{h}^{(t)}$ (see Appendix E.1). The b-node embedding is computed as $e_{b,u}^{(t_i)} = \tanh\big(W_b \cdot (\tanh\big(f^{(a,t_i)}\big) \odot e_{\text{tl}}^{(t_i)})\big)$.

**Geometric step decomposition.** Finally, Walk2Pers models preference evolution as a directed, structured geometric step, balancing persistence with novelty $\Phi(\cdot)$:

$$\mathbf{e}^{\mathbf{c}_{b,u}^{(t)}} = e_{b,u}^{(t)} + \text{mag}^{(t)}\Big( \cos\theta^{(t)} \cdot u^{(t-1)} + \sin\theta^{(t)} \cdot o^{(t)} \Big), \tag{5}$$

where $u^{(t-1)}$ is the momentum axis (continuity), $o^{(t)}$ its orthogonal novelty axis, $\text{mag}^{(t)}$ the step size, and $\theta^{(t)}$ the rotation angle interpolating between persistence ($\theta \approx 0$) and novelty ($\theta$ large). This realizes the $\Phi$ term in & Eq. 3 (Implementation & theoretical equivalence: Appendices G G.2).

**Training.** The encoder is supervised via two complementary objectives. First, the *next-node prediction head* maps the final contextualized embedding to the predicted query b-node: $\mathbf{e}_{b_q,u}^{(t+1)} = \mathbf{W}_{\text{next}} \cdot \mathbf{e}_{b,u}^{\mathbf{c}(t)} + b_{\text{next}}$. Second, a *position classifier* enforces alignment between the contextualized trajectory and its constituent steps: $\hat{\mathbf{p}}_b^{(t)} = \text{SoftMax}(W_{\text{pos}} \cdot \mathbf{e}_{b,u}^{\mathbf{c}(t)})$. The joint objective combines these two signals: $\mathcal{L}_{\text{align}} = -\frac{1}{l} \sum_{i=1}^{l} \log \hat{p}_b^{(t_i)}; \mathcal{L}_{\text{next}} = -\log \hat{p}_b^{(t)}; \mathcal{L}_{\text{enc}} = \alpha \, \mathcal{L}_{\text{align}} + (1-\alpha) \, \mathcal{L}_{\text{next}};$ $\alpha = 0.6$ so as to avoid cumulative cascading of $\mathcal{L}_{\text{align}}$ on $\mathcal{L}_{\text{next}}$. The alignment term ensures that each intermediate b-node in the trajectory is recoverable from the contextualized embedding, regularizing the walk so it respects positional consistency across steps. The next-node prediction term directly trains the encoder to forecast the upcoming behavior node, making the walk predictive rather than descriptive. Together, these losses encourage $\Phi$ (geometric step), $\Psi$ (dual memories), and $\Delta$ (drift) to cooperate in producing embeddings that are both history-faithful and forward-looking.

### 3.3.2 DECODER: CONTEXTUALIZING USER INTENT FOR SUMMARIZATION

While the encoder (Sec. 3.3.1) is the core instantiation of the Structured Walk Hypothesis, we attach the same backbone decoder of the encoder-decoder model used to generate seed embeddings, T5-base, that consumes the contextualized query embedding $\mathbf{e}_{d_q,u}^c$ and generates the personalized summary. Training optimizes a combined objective: $\mathcal{L}_{\text{dec}} = \text{Average}(\mathcal{L}_{\text{gen}}, \mathcal{L}_{\text{enc}})$, where $\mathcal{L}_{\text{gen}}$ is cross-entropy under teacher forcing and $\mathcal{L}_{\text{enc}}$ is the structured-walk encoder loss from Sec. 3.3.1. This ensures that the encoder faithfully models user trajectories while the decoder exploits those states to produce preference-aware summaries. We evaluate two decoder variants: **T5-CA (Contextualized Attention)** and **T5-UCA (User-aware CA)**. The T5-CA-Decoder contextualizes the query document embedding $\mathbf{e}_{d_q}$ with the latent summary intention vector (s-node) via cross-attention. This injects *summary intent* but leaves the document representation agnostic of the user's history. T5-UCA-Decoder builds on that by gating the query document embedding with the user's trajectory state $\mathbf{e}_{q,b_{u_j}}^{(t_l)}$. The gating suppresses aspects aligned with $h^-$ (e.g., topics repeatedly skipped) and amplifies aspects aligned with $h^+$ (reinforced interests), producing a user-aware document vector. This ensures that the same document is viewed through Alice's preference lens differently than through Bob's. The gated representation is then contextualized with the latent summary intent, as in CA. While CA adapts summaries to "*what this document is generally about given the latent summary signal*", UCA further adapts to "*what this document means for **this user** given their interaction history.*" As shown in Sec. 5.2, UCA yields stronger personalization, while CA serves as a weaker control. Full derivations of latent s-node extraction and gating functions are in Appendix G.3.

## 4 EVALUATION

We design the experiments to address the following research questions (RQ): **RQ1:** Is MDH sufficient for modeling preference evolution in summarization? **RQ2:** Do the necessary components of the SWH, i.e., dual memory lanes + summary-specific drift & action-conditioned geometric steps with magnitude–orientation decomposition, yield systematic gains over MDH variants? **RQ3:** How does `Walk2Pers`, as an instantiation of SWH, compare against specialized personalized summarizers, prompt-personalized LLMs, and oracle summarizers?

### 4.1 EXPERIMENT SETUP

**Training (& Test) Datasets.** We evaluate across three corpora capturing diverse personalization signals: (i) **PENS** (Ao et al., 2021), a large-scale news summarization dataset with user clicks/skips; (ii) **PersonalSum (EN)** (Zhang et al., 2024), a manually curated dataset translated to English with explicit summary requests; (iii) **OpenAI-Reddit** (Völske et al., 2017), where summaries are user-rated and span diverse domains. We construct UIGs for each dataset $\mathcal{T}_{\text{train}}$ and slice trajectories before every ($d$–$s$) pair, yielding history $\tau_h^{u_j}$, query document $d_q$, and target summary $s_{q,u_j}^*$. For PENS ($\mathcal{T}_{\text{train}}^{\text{PENS-D}}$), we sample 55K training trajectories (avg. 134 $d$-nodes, 5 $s$-nodes per trajectory). For OpenAI-Reddit ($\mathcal{T}_{\text{train}}^{\text{OAI}}$), 18K training trajectories (avg. 39 $d$-nodes, 10 $s$-nodes per trajectory). For PersonalSum-EN ($\mathcal{T}_{\text{train}}^{\text{PS-EN}}$) (translated into English using M2M-100 (Fan et al., 2020)), 700 trajec-

tories (highly curated, long-horizon). The corresponding Test sets ($\mathcal{T}_{\text{test}}$) reflect the same structure, with additional random skips injected (50–70 per user in PENS) to stress-test suppression memory[1].

**Training Setup.** `Walk2Pers` is trained end-to-end with the joint encoder–decoder objective $\mathcal{L}_{\text{dec}}$ for 6 epochs. Then the encoder is frozen while the decoder (including the last 6 layers of the T5 decoder) is further finetuned for 18 epochs. This ensures encoder quality is the primary driver of downstream summarization gains. Training details are in Appendix H.2 and Table 13.

## 4.2 BASELINES

**A. Encoder Baselines**   We benchmark `Walk2Pers` for **RQ-1 & 2** against five MDH models:

**I. Short-Memory Drift (SMD).** This minimal baseline captures the pure MDH stance: each new b-node embedding is computed only from the immediate past state and the current action embedding. No reinforcement, suppression, or geometric structure is retained. User preference evolution reduces to a one-step drift, overwriting longer histories. Details in Appendix E.1.

**II. Action-Gated Drift (AGD).** This stronger MDH variant replaces the update rule of SMD with action-specific gates. Different parameterizations are applied for `click`, `skip`, `genSumm`, and `summGen`, allowing the model to qualitatively differentiate between actions. However, the updates remain short-memory: past interactions vanish quickly, and no persistent reinforcement or suppression is maintained. Details in Appendix E.1.

**III. Short-Memory Neural-Encoders.** We also include **NAML** (Wu et al., 2019a), **NRMS** (Wu et al., 2019b), and **EBNR** (Okura et al., 2017), widely used in personalized recommendation and summarization pipelines (e.g., PENS). These models differ in mechanism. NAML employs additive self-attention over multi-view document features, NRMS uses multi-head self-attention, and EBNR leverages a GRU over clicked entities. However, all collapse long histories into compressed short-memory embeddings, consistent with MDH.

These baselines constitute strong MDH realizations for the **next-b-node prediction task**. The candidate set contains 151 b-nodes (including the target). We report average AUC, MRR, and nDCG@5/10 over the PENS test set (metric definitions in Appendix B), with full baseline formalizations in Appendix E.1. These metrics directly measure a model's ability to predict real user behavior because each ground-truth b-node corresponds to an actual user action (`click`, `skip`, `summarize`) and, for <sumGen, s-node> pairs, to a **human-written summary**. Hence, RQ1/2 evaluate *behavioral faithfulness*.

**B. Personalized Summarizers** To validate the efficacy of `Walk2Pers` in terms of the downstream personalized summarization task, we also compare against SOTA personalized summarizers.

**I. Neural-Encoder Augmented Summarizers.** For **RQ-3**, we compare against three SOTA personalized summarizers: PENS (Ao et al., 2021), GTP (Song et al., 2023), and Signature-Phrase (Cai et al., 2023). PENS pairs a pointer generator with external user encoders; GTP integrates the TrRMIo encoder internally. Within PENS, we use NAML (Wu et al., 2019a) (T-1), EBNR (Okura et al., 2017) (T-1), and NRMS (Wu et al., 2019b) (T-2), with injection (T-x) details in Appendix E.2.1. TrRMIo is an integrated full-sequence Transformer, while Signature-Phrase models user-specific keyphrases. All baselines are finetuned on $\mathcal{T}_{\text{train}}^{\text{P}}$ under the same regime as `Walk2Pers`.

**II. LLMs-as-Summarizers.**   To extend **RQ-3**, we evaluate six **frozen** instruction-tuned LLMs—Gemini-2.5-Flash, Qwen3-235B (Team, 2025), Mistral-7B-Instruct (Jiang et al., 2023), DeepSeek-R1-14B (DeepSeek-AI et al., 2025), LLaMA-2-13B-Chat-HF (Touvron et al., 2023), and Zephyr-7B (Tunstall et al., 2023). Gemini and Qwen are included for their SOTA LongBench performance. We use the best 0-shot and 2-shot prompting recipes from Patel et al. (2024) and apply prompt-chaining for DeepSeek-R1-14B and Mistral-7B-Instruct. Prompt formats are in Appendix J.

**III. Non-Personalized Summarizers as Oracles.** As part of **RQ-3**, we evaluate three strong non-personalized summarizers, BigBird-Pegasus (Zaheer et al., 2020), SimCLS (Liu & Liu, 2021), and T5-base (Raffel et al., 2020), under the "oracle" protocol of Vansh et al. (2023). We replace each query document's title with the user's gold-reference title to inject the true preference cue. This assesses whether frozen models can exploit the cues to produce seemingly personalized outputs.

---

[1]For detailed stress test results see Appendix I.2 and I.3.

Table 2: **RQ-1/2: Next b-node Prediction (PENS Dataset)**: Details in Table 14; Full result for **Cross-task transferability** on Sequential News Recommendation (MIND Dataset): Table 17.

| Category | Models | AUC | MRR | nDCG@5 | nDCG@10 |
|---|---|---|---|---|---|
| MDH based | NAML | 0.498 | 0.001 | 0.0004 | 0.0007 |
| | NRMS | *0.499* | 0.0009 | 0.0002 | 0.0004 |
| | EBNR | *0.499* | 0.0009 | 0.0003 | 0.0005 |
| | SMD (ours) | 0.415 | 0.094 | 0.052 | 0.065 |
| | AGD (ours) | 0.446 | 0.113 | 0.069 | 0.073 |
| SWH based | `Walk2Pers`-Encoder w/o Geometric Step (ours) | 0.474 | *0.121* | *0.082* | *0.132* |
| | `Walk2Pers`-Encoder Full (ours) | **0.532** | **0.23** | **0.198** | **0.249** |

Table 3: **(RQ-3:) `Walk2Pers`-Encoder Validation:** Personalized summarization performance (w.r.t PSE-JSD/SU-4/METEOR) comparison with SOTA baselines on the PENS dataset.

| Category | Model | PSE-JSD | PSE-SU4 | PSE-METEOR |
|---|---|---|---|---|
| Oracle Summarizers (via Cue Injection) | BigbirdPegasus | 0.253 | 0.143 | 0.168 |
| | SimCLS | 0.157 | 0.032 | 0.116 |
| | T5-base | 0.073 | 0.011 | 0.022 |
| LLMs (w/ 0-shot user history) | LLaMA-13B | 0.187 | 0.069 | 0.078 |
| | Zephyr-7B | 0.211 | 0.081 | 0.089 |
| | Mistral-7B | 0.212 | 0.082 | 0.098 |
| | DeepSeek-14B | 0.152 | 0.078 | 0.084 |
| LLMs (w/ 2-shot user history) | LLaMA-13B | 0.227 | 0.078 | 0.081 |
| | Zephyr-7B | 0.231 | 0.085 | 0.086 |
| | Mistral-7B | 0.235 | 0.087 | 0.084 |
| | DeepSeek-14B | 0.248 | 0.094 | 0.097 |
| | Qwen-3-235B | 0.105 | 0.082 | 0.082 |
| | Gemini-2.5-Flash | 0.222 | 0.104 | 0.124 |
| LLMs (Prompt-chaining) | Mistral-7B | 0.072 | 0.026 | 0.023 |
| | DeepSeek-14B | 0.078 | 0.028 | 0.024 |
| Fine-tuned Specialized (Personalized) $\sim$ MDH | PENS-NAML-T1 | 0.021 | 0.014 | 0.016 |
| | PENS-EBNR-T1 | 0.015 | 0.010 | 0.011 |
| | PENS-EBNR-T2 | 0.011 | 0.008 | 0.009 |
| | PENS-NRMS-T1 | 0.015 | 0.011 | 0.011 |
| | PENS-NRMS-T2 | 0.008 | 0.007 | 0.007 |
| | GTP | 0.024 | 0.017 | 0.019 |
| | SP-Individual | 0.017 | 0.015 | 0.014 |
| Markov Drift (MDH) Encoders (ours) | SMD + T5-UCA-Decoder | 0.143 | 0.136 | 0.107 |
| | AGD + T5-UCA-Decoder | 0.286 | 0.214 | 0.248 |
| `Walk2Pers`-Encoders ($\sim$ SWH) (ours) | – w/o Geometric Step + T5-UCA-Decoder | 0.306 | 0.334 | 0.321 |
| | – Full + T5-CA-Decoder | *0.418* | *0.341* | *0.422* |
| | – Full + T5-UCA-Decoder | **0.452** | **0.383** | **0.449** |

## 5 RESULTS AND OBSERVATIONS

We present results w.r.t the three RQs (Section 4). All results have a significance of $p < 0.05$.

### 5.1 WALK2PERS-ENCODER PREDICTION ACCURACY

**RQ1 — Are MDH models sufficient?** We observe that MDH baselines fail to anticipate user trajectories reliably. Short-memory neural encoders (NAML/NRMS/EBNR) hover at chance AUC ($\approx 0.5$) and collapse on rank metrics (MRR $\leq 0.001$, nDCG $\leq 5 \times 10^{-4}$), showing that compressed hidden states carry little predictive signal. Among controlled variants, action gating helps (AGD > SMD, e.g., MRR 0.113 vs. 0.094), but overall accuracy remains low, as past signals vanish quickly and reinforcement/suppression is absent.

**RQ2 — Do SWH components yield systematic gains?** We find that adding dual memories and drift (`Walk2Pers`-Enc. w/o Geo) already surpasses the best baseline AGD (AUC $+0.028$, nDCG@10 $+0.059$). Geometric magnitude–orientation step yields large additional jumps (AGD: AUC $+0.086/ + 0.117/ + 0.176$ w.r.t AUC/MRR/nDCG@5/10). These results confirm that memory lanes preserve reinforcement/suppression, while geometric steps capture continuity vs. novelty.

**Cross-task Performance.** As a further validation of `Walk2Pers` encoder, we evaluate the sequential recommendation performance of it on widely adopted MIND news recommendation dataset. We find `Walk2Pers` encoder (Full, with geometric step), which was trained end-to-end for personalized summarization task, to surpass the MIND recommendation leaderboard baselines by a significant margin. It outperforms the best baseline (Fastformer+PLM-NR-Ensemble) by 1.2↑ on MRR, 1.8↑ on nDCG@5, and 3.5↑ on nDCG@10. This result demonstrates the cross-task transferability of `Walk2Pers` encoder. Detailed results are in Table 17.

Table 4: **(RQ-3) Cross-domain generalizability on OpenAI (Reddit)**

| Category | Model | PSE-JSD | PSE-SU4 | PSE-METEOR |
|----------|-------|---------|---------|------------|
| LLMs (w/ 2-shot user history) | LLaMA-13B | 0.232 | 0.093 | 0.107 |
| | Zephyr-7B | 0.214 | 0.087 | 0.104 |
| | Mistral-7B | 0.226 | 0.088 | 0.103 |
| | DeepSeek-14B | _0.243_ | _0.095_ | _0.109_ |
| **Walk2Pers** | Full + T5-UCA-Decoder | **0.339** | **0.303** | **0.350** |

## 5.2 RQ-3: WALK2PERS END-TO-END SUMMARIZATION PERFORMANCE

To address **RQ3**, we benchmark `Walk2Pers` against the strong summarizer baselines as described in Section 4.2. Evaluation is conducted on PENS and OpenAI (Reddit) datasets and further validated using the PersonalSumm dataset using the three PerSEval metrics (PSE-JSD/SU4/METEOR), which correlate strongly with human preference ratings (Appendix A.2.1).

**Comparison with Specialized Personalized Summarizers.** We see that `Walk2Pers` consistently outperform all specialized models (Table 3). While SOTA frameworks like GTP and SP yield PSE scores in the $0.017$–$0.024$ range, `Walk2Pers` achieves $0.452/0.383/0.449$ (JSD/SU4/METEOR), corresponding to average absolute gains of $0.41$ over these fine-tuned MDH instantiations. This highlights that explicit geometric-step modeling with dual memory lanes captures preference evolution more effectively than both RNN- and Transformer-based encoders.

**Comparison with LLMs.** LLMs improve when conditioned with user histories. However, they remain substantially below `Walk2Pers`. For instance, the best two-shot configuration (DeepSeek-14B) is outperformed by `Walk2Pers` by margins of $0.20/0.29/0.35$. On average, `Walk2Pers` yields gains of $0.22$ across all LLMs (including long context models Gemini and Qwen3). Prompt-chaining is even less effective, falling behind even MDH baselines. Furthermore, evaluation on the **OpenAI (Reddit)** dataset (Table 4) confirms that `Walk2Pers` **indicates cross-domain generalization**, outperforming the strongest LLM baseline by $0.09/0.20/0.24$ across JSD/SU4/METEOR. We also evaluate `Walk2Pers` on the translated PersonalSum dataset and observe scores of $0.31/0.28/0.3$ w.r.t. PSE-JSD/SU4/METEOR, underscoring the efficacy in datasets tailored for personalized summarization. These results suggest that while prompting improves LLMs, it cannot substitute a dedicated action-aware encoder that explicitly models user trajectories.

**Comparison with Oracle Summarizers.** We observe that oracle summarizers, augmented with gold preference cues, fall short. BigBird-Pegasus, the strongest oracle, is outperformed by a margin of $0.20/0.24/0.28$. Relative to SimCLS and T5-base, the margins are even larger ($0.29/0.35/0.43$ and $0.43/0.37/0.38$, resp.). This underscores that `Walk2Pers` utilizes evolving preferences better than the best-possible performance of non-personalized architectures.

**Performance w.r.t. Accuracy.** We evaluate the accuracy of `Walk2Pers` w.r.t. gold-reference summaries under standard accuracy evaluation metrics Rouge-L and Rouge-SU4 (Lin, 2004), and find that `Walk2Pers` outperforms the baselines by an average margin of $22.3/24.1$ on RG-L and RG-SU4, respectively. This confirms that a boost in personalization capability also bridges the lack of accuracy gap. Results are in Table 18.

## 5.3 HUMAN-RATING GROUNDED EVALUATION

While RQ1/2 provide behavioral faithfulness against human trajectories, as a part of RQ-3, we complement this by assessing how generated summaries align with what users *prefer*. Using the multi-domain non-news OpenAI-Reddit dataset, which contains multiple (5781) human-rated summaries of 9 models for 642 query documents, we identify 1042 top-rated (i.e., 7) one per user as the *human-preferred reference*. We then measure the SBert-embedding-space RMSD-divergence of the model-generated summaries from the reference and create a ground rating-to-RMSD-range map table, where each rating row has its corresponding average min-max range. Using this table, we interpolate the HJ-rating of our baseline models as in Table 5. We observe that both full and w/o geometric step variants (trained on OpenAI-train) of `Walk2Pers` achieve an average rating of 7 out of 7, while MDH-models are significantly underperforming, with the exception of AGD. Interpolation computation details in Appendix I.1.

Table 5: **Personalized Summarization Performance w.r.t Human-Judgment Ratings:** Avg. interpolated rating (w/ RMSD dist. from gold reference) on OpenAI (Reddit) dataset

| Category | Model | RMSD (generated vs. gold reference) | HJ Rating |
|---|---|---|---|
| Fine-tuned Specialized (Personalized) ∼ MDH | PENS+EBNR-T1 | 0.932 | 2 |
| | PENS+EBNR-T2 | 0.938 | 2 |
| | PENS+NAML-T1 | 0.926 | 2 |
| | PENS+NRMS-T1 | 0.911 | 2 |
| | PENS+NRMS-T2 | 0.919 | 2 |
| | GTP+TrRMIo | 0.939 | 2 |
| | SP | 0.881 | 3 |
| Best-forming LLMs (w/ 2-shot user history) | Mistral | 0.791 | 5 |
| | Gemini | 0.782 | 5 |
| | DeepSeek | 0.779 | 5 |
| T5-UCA-Decoder + MDH Encoders | SMD | 0.836 | 4 |
| | AGD | 0.701 | _6_ |
| SWH w/ T5-UCA-Decoder | Walk2Pers w/o Geometric Step | _0.461_ | 7 |
| | Walk2Pers (Full) | **0.396** | 7 |

## 6 CONCLUSION

In this paper, we contrast the *Markovian Drift Hypothesis* (MDH) with our proposed *Structured Walk Hypothesis* (SWH), which models preference evolution as an action-conditioned geometric walk with memory. Walk2Pers, our proposed instantiation of SWH, encodes trajectories through action-aware b-cells, dual memory lanes, and magnitude–orientation steps, balancing continuity with novelty. Experiments show that the next behavior (b-node) prediction, the primary test of SWH, outperforms MDH baselines. At the same time, downstream personalized summarization confirms that stronger preference modeling yields more user-aligned outputs. While limited by fixed memory kernels and reliance on limited noisy logs, Walk2Pers demonstrates that preference evolution is better framed as SWH than as shallow MDH.

## CODE OF ETHICS STATEMENT

This research adheres to the ICLR Code of Ethics[2]. In conducting this work, we: (i) contributed to society and human well-being by advancing methods for trustworthy personalized summarization; (ii) upheld high standards of scientific excellence through transparent reporting, reproducibility, and acknowledgement of prior work; (iii) avoided harm by ensuring that our methods were tested responsibly, with no foreseeable misuse to compromise safety, security, or privacy; (iv) were honest, trustworthy, and transparent in disclosing our methods, limitations, and potential risks; (v) acted fairly and without discrimination, considering inclusivity in data and evaluation; (vi) respected the work and rights of others via proper citation and intellectual property compliance; and (vii) respected privacy by not using personally identifiable or sensitive information in our datasets. (viii) used LLMs (GPT-5) limited to structural changes (paraphrasing and summarization of our own content, which has not been used verbatim in most of the paper), table format corrections, and extensive literature review (using Deep Research). We have not used LLM for any content *generation* purpose.

## REPRODUCIBILITY STATEMENT

We have made significant efforts to ensure the reproducibility of our work. All details of the proposed Walk2Pers framework, including the encoder and decoder variants, training objectives, and evaluation protocols, are provided in Sections 3.3 and 4. Hyperparameter choices, model configurations, and training details are documented in Appendix H.2 and Table 13. Dataset descriptions, preprocessing steps, and evaluation metrics across personalized summarization (PSE-JSD, PSE-SU-4, PSE-METEOR), sequential predictions and next b-node prediction (nDCG, MRR) are clearly specified in Section 2, Appendices C, A.2.1, and B. We detail the human judgment annotations in Appendix I.1. We also provide ablation studies (Tables 2 & 3) to demonstrate robustness to design choices. To facilitate independent verification, we include the corresponding GitHub code repository: https://github.com/KDM-LAB/Walk2Pers-ICLR-2026.git.

---

[2] https://iclr.cc/public/CodeOfEthics

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

# A  MEASURING DEGREE-OF-PERSONALIZATION

## A.1  MOTIVATION

Vansh et al. (2023) proposed EGISES– a metric to measure the degree of **in**sensitivity-to-subjectivity for relative benchmarking of how much models *lack personalization* (i.e., a lower score is better within the range $[0, 1]$) instead of assigning an absolute goodness score. Based on this notion, they defined (summary-level) "**deviation**" of a model $M_{\theta, u}$(later termed as ***Degree-of-Responsiveness*** (DEGRESS) by Dasgupta et al. (2024)) as follows:

**Summary-level DEGRESS.** Given a document $d_i$ and a user-profile $u_{ij}$ (user $j$'s expected summary), the summary-level responsiveness of a personalized model $M_{\theta, u}$, (i.e., $\text{DEGRESS}(s_{u_{ij}}|(d_i, u_{ij}))$), is defined as the *proportional* divergence between model-generated summary $s_{u_{ij}}$ of $d_i$ for $j$-th user from other user-specific summary versions w.r.t a corresponding divergence of $u_{ij}$ from the other user-profiles.

$\text{DEGRESS}(s_{u_{ij}}|(d_i, u_{ij}))$ is formulated as:

$$\text{DEGRESS}(s_{u_{ij}}|(d_i, u_{ij})) = \frac{1}{|\mathbf{U}_{d_i}|} \sum_{k=1}^{|\mathbf{U}_{d_i}|} \frac{min(X_{ijk}, Y_{ijk}) + \epsilon}{max(X_{ijk}, Y_{ijk}) + \epsilon}$$

$$X_{ijk} = \frac{\exp(w(u_{ij}|u_{ik}))}{\sum\limits_{l=1}^{|\mathbf{U}_{d_i}|} \exp(w(u_{ij}|u_{il}))} \cdot \sigma(u_{ij}, u_{ik}); \quad Y_{ijk} = \frac{\exp(w(s_{u_{ij}}|s_{u_{ik}}))}{\sum\limits_{l=1}^{|\mathbf{U}_{d_i}|} \exp(w(s_{u_{ij}}|s_{u_{il}}))} \cdot \sigma(s_{u_{ij}}, s_{u_{ik}}) \tag{6}$$

$$w(u_{ij}|u_{ik}) = \frac{\sigma(u_{ij}, u_{ik})}{\sigma(u_{ij}, d_i)}; \quad w(s_{u_{ij}}|s_{u_{ik}}) = \frac{\sigma(s_{u_{ij}}, s_{u_{ik}})}{\sigma(s_{u_{ij}}, d_i)}$$

Here, $|\mathbf{D}|$ is the total number of documents in the evaluation dataset, $|\mathbf{U}|$ is the total number of users who created gold-reference summaries that reflect their expected summaries (and thereby, their subjective preferences), and $|\mathbf{U}_{d_i}| (= |\mathbf{S}_{d_i}|)$ is the number of users who created gold-references for document $d_i$. $w$ is the divergence of the model-generated summary $s_{u_{ij}}$ (and the corresponding expected summary $u_{ij}$) from document $d_i$ itself in comparison to all the other versions. It helps to determine how much percentage (therefore, the softmax function) of the divergence (i.e., $\sigma(s_{u_{ij}}, s_{u_{ik}})$ should be considered for the calculation of DEGRESS. If $s_{u_{ij}}$ is farther than $s_{u_{ik}}$ w.r.t $d_i$ then $\text{DEGRESS}(s_{u_{ij}}|(d_i, u_{ij})) < \text{DEGRESS}(s_{u_{ik}}|(d_i, u_{ik}))$, implying that $M_{\theta, u}$ is more responsive to the $k$-th reader. A lower value of $\text{DEGRESS}(s_{u_{ij}}|(d_i, u_{ij}))$ indicates that while reader-profiles

are different, the generated summary $s_{u_{ij}}$ is very similar to other reader-specific summaries (or vice versa), and hence, is not responsive at the summary-level. The system-level DEGRESS and EGISES have been formulated as follows:

$$\text{DEGRESS}(M_{\boldsymbol{\theta},u}) = \frac{\sum_{i=1}^{|\mathbf{D}|} \frac{\sum_{j=1}^{|\mathbf{U}_{d_i}|} \text{DEGRESS}(s_{u_{ij}}|(d_i,u_{ij}))}{|\mathbf{U}_{d_i}|}}{|\mathbf{D}|} \tag{7}$$

## A.2 PERSEVAL: FORMULATION

As can be noted, the **DEGRESS formualtion does not enforce any penalty on accuracy drop**. To rectify this Dasgupta et al. (2024) proposed PerSEval. The design of PerSEval had two key goals: (i) to penalize models for poor accuracy, while simultaneously (ii) ensuring that the evaluation of responsiveness (i.e., DEGRESS) is not overshadowed by high accuracy. This penalty is referred to as the *Effective DEGRESS Penalty Factor* (EDP). If a model achieves 100% accuracy, no EDP will be applied, and the PerSEval score will equal the DEGRESS score. The following formulatiown of PerSEval guarantees these properties:

$$\text{PerSEval}(s_{u_{ij}}|(d_i,u_{ij})) = \text{DEGRESS}(s_{u_{ij}}|(d_i,u_{ij})) \times \text{EDP}(s_{u_{ij}}|(d_i,u_{ij}))$$

$$\text{where, } \text{EDP}(s_{u_{ij}}|(d_i,u_{ij})) = 1 - \frac{1}{1 + 10^{\alpha \geq 3} \cdot \exp\left(-(10^{\beta \geq 1} \cdot \text{DGP}(s_{u_{ij}}|(d_i,u_{ij})))\right)}, \tag{8}$$

$$\text{DGP}(s_{u_{ij}}|(d_i,u_{ij})) = \text{ADP}(s_{u_{i*}}|(d_i,u_{i*})) + \text{ACP}(s_{u_{ij}}|(d_i,u_{ij}))$$

Here, ADP is a document-level penalty due to a drop in accuracy for the best-performance of the model (i.e., the model-generated summary of document $d_i$ ($s_{u_{ij}}$) is closest to the corresponding reader's expected summary $u_{ij}$). ADP is formulated as follows:

$$\text{ADP}(s_{u_{i*}}|(d_i,u_{i*})) = \frac{1}{1 + 10^{\gamma \geq 4} \cdot \exp\left(-10 \cdot \frac{\sigma^*(s_{u_{i\bullet}},u_{i\bullet})|d_i - \mathbf{0}}{(\mathbf{1} - \sigma^*(s_{u_{i\bullet}},u_{i\bullet})|d_i) + \epsilon}\right)}$$

$$\text{where, } \sigma^*(s_{u_{i\bullet}},u_{i\bullet})|d_i = \min_{j=1}^{|\mathbf{U}_{d_i}|} \sigma(s_{u_{ij}},u_{ij})|d_i \tag{9}$$

$$\text{and } \{\epsilon : \text{An infinitesimally small number} \in (0,1)\}$$

ADP ensures that even if the DEGRESS score is acceptable, a penalty due to accuracy drop can still be imposed as a part of EDP. ADP, however, fails to address the scenario where the best-case scenario is acceptable (i.e., accuracy is fairly high) but is rather an outlier case – i.e., for most of the other model-generated summary versions, there is a considerable accuracy drop. To address this issue, the second penalty component within EDP called *Accuracy-inconsistency Penalty* (ACP) was introduced which evaluates whether a model consistently performs w.r.t accuracy for a specific generated summary compared to its average performance. ACP is formulated as:

$$\text{ACP}(s_{u_{ij}}|(d_i,u_{ij})) = \frac{1}{1 + 10^{\gamma \geq 4} \cdot \exp\left(-10 \cdot \frac{\sigma(s_{u_{ij}},u_{ij})|d_i - \sigma^*(s_{u_{i\bullet}},u_{i\bullet})|d_i}{(\overline{\sigma}(s_{u_{i\bullet}},u_{i\bullet})|d_i - \sigma^*(s_{u_{i\bullet}},u_{i\bullet})|d_i) + \epsilon}\right)}$$

$$\text{where, } \overline{\sigma}(s_{u_{i\bullet}},u_{i\bullet})|d_i = \frac{1}{|\mathbf{U}_{d_i}|} \sum_{j=1}^{|\mathbf{U}_{d_i}|} \sigma(s_{u_{ij}},u_{ij})|d_i \tag{10}$$

The system-level PerSEval score is as follows:

$$\text{PerSEval}(M_{\boldsymbol{\theta},u}) = \frac{\sum_{i=1}^{|\mathbf{D}|} \frac{\sum_{j=1}^{|\mathbf{U}_{d_i}|} \text{PerSEval}(s_{u_{ij}}|(d_i,u_{ij}))}{|\mathbf{U}_{d_i}|}}{|\mathbf{D}|} \tag{11}$$

The system-level PerSEval $\in [0,1]$ and is bounded by the system-level DEGRESS score.

### A.2.1  PSE METRICS

**PerSEval-RG-SU4**   (or PSE-SU4) is the `PerSEval` variant that uses ROUGE-SU4 (Lin, 2004) as a distance metric (i.e., $\sigma$) in the `PerSEval` formula. PSE-SU4 has been reported to have high human-judgment correlation (Pearson's $r$: 0.6; Spearman's $\rho$: 0.6; Kendall's $\tau$: 0.51) Dasgupta et al. (2024). The **ROUGE-SU4** score is based on *skip-bigrams*, which are pairs of words that appear in the same order within a sentence but can have up to four other words between them. The formula is as follows:

For a given generated summary $G$ and reference summary $R$, the ROUGE-SU4 score is calculated as:

**Skip-Bigram Recall ($R_{SU4}$):**

$$R_{\text{SU4}} = \frac{\text{Count of matching skip-bigrams between } G \text{ and } R}{\text{Total skip-bigrams in } R}$$

**Skip-Bigram Precision ($P_{SU4}$):**

$$P_{\text{SU4}} = \frac{\text{Count of matching skip-bigrams between } G \text{ and } R}{\text{Total skip-bigrams in } G}$$

**F1 Score ($F1_{SU4}$):** The F1 score is the harmonic mean of precision and recall:

$$F1_{\text{SU4}} = \frac{2 \times P_{\text{SU4}} \times R_{\text{SU4}}}{P_{\text{SU4}} + R_{\text{SU4}}}$$

Where:

- A **skip-bigram** consists of two words in the correct order but with zero to four words skipped in between.
- Matching skip-bigrams are counted between the generated summary and the reference summary.

The final **ROUGE-SU4** score is typically reported as the F1 measure, balancing precision and recall.

**PerSEval-JSD**   (or PSE-JSD) is the `PerSEval` variant that uses the Jensen–Shannon Divergence (JSD) (Menéndez et al., 1997) as the distance metric $\sigma$ in the `PerSEval` formula. JSD is a smoothed and symmetric version of Kullback–Leibler divergence between the unigram (or n-gram) distributions of the generated summary $G$ and reference summary $R$. Its formulation is:

$$\text{JSD}(P \,\|\, Q) \;=\; \tfrac{1}{2}\,\text{KL}\big(P \,\|\, M\big) \;+\; \tfrac{1}{2}\,\text{KL}\big(Q \,\|\, M\big) \quad \text{where} \quad M = \tfrac{1}{2}(P + Q) \tag{12}$$

here, $P$ and $Q$ are the normalized n-gram probability distributions of $G$ and $R$ respectively, and

$$\text{KL}(P\|M) \;=\; \sum_{x} P(x) \log \frac{P(x)}{M(x)}.$$

We then define the divergence as: $\sigma_{\text{JSD}}(G, R) = \text{JSD}\big(P_G \| P_R\big)$ and plug $\sigma_{\text{JSD}}$ into all occurrences of $\sigma$ in Equations equation 6–equation 11 to obtain PSE-JSD.

**PerSEval-Meteor**   (or PSE-Meteor) uses the METEOR score Banerjee & Lavie (2005); Lavie & Agarwal (2007) as the similarity metric; we convert it into a distance by $1 - \text{METEOR}$. METEOR aligns unigrams (with synonymy, stem, and paraphrase matching) and combines precision, recall, and a fragmentation penalty. Its formulation is:

$$P = \frac{|\text{matched\_unigrams}|}{|\text{unigrams}(G)|}, \quad R = \frac{|\text{matched\_unigrams}|}{|\text{unigrams}(R)|}, \tag{13}$$

$$F_\alpha = \frac{P\,R}{\alpha\,P + (1-\alpha)\,R}, \quad \alpha \in [0,1], \tag{14}$$

$$\text{Penalty} = \gamma \left( \frac{\#\text{chunks}}{|\text{matched\_unigrams}|} \right)^\beta, \quad \gamma, \beta > 0, \tag{15}$$

$$\text{METEOR}(G, R) = (1 - \text{Penalty}) \times F_\alpha. \tag{16}$$

We then set $\sigma_{\text{Meteor}}(G, R) = 1 - \text{METEOR}(G, R)$, and substitute $\sigma_{\text{Meteor}}$ for $\sigma$ in Equations equation 6–equation 11 to yield PSE-Meteor.

## B PREDICTION METRICS

In this section, we provide definitions of the evaluation metrics used in our experiments: *AUC*, *MRR*, and *nDCG@k*. Each metric captures a complementary aspect of ranking quality when comparing the predicted next-step positions against the ground-truth target.

**Area Under Curve (AUC)**   AUC measures how well the model ranks the ground-truth item relative to all other candidates. Formally, if $r$ is the rank (1-based) of the ground-truth item among $C$ candidates, we define:

$$\text{AUC} = \frac{C - r}{C - 1}.$$

This normalizes the rank to the interval $[0, 1]$, where higher values indicate that the true item is ranked closer to the top. Intuitively, AUC reflects the overall discriminative ability of the model.

**Mean Reciprocal Rank (MRR)**   MRR emphasizes how highly the correct item appears in the ranked list. Given the rank $r$ of the ground-truth item, its reciprocal rank is:

$$\text{RR} = \frac{1}{r}.$$

MRR is the average of RR values across all queries. MRR rewards systems that consistently place the true item very close to the top of the ranking.

**Normalized Discounted Cumulative Gain (nDCG@k)**   nDCG@k evaluates the quality of the top-$k$ predictions, with stronger weight on higher-ranked positions. The gain of a relevant item at rank $r$ is discounted logarithmically:

$$\text{DCG@}k = \sum_{i=1}^{k} \frac{\mathbb{I}\{r_i = \text{target}\}}{\log_2(1 + i)},$$

where $\mathbb{I}\{\cdot\}$ is an indicator function. Since there is only one relevant target per query, DCG@$k$ reduces to $\frac{1}{\log_2(1+r)}$ if the target appears within the top-$k$, and 0 otherwise. Normalization divides by the best possible score (which is 1 if the target is at rank 1). Thus:

$$\text{nDCG@}k = \begin{cases} \frac{1}{\log_2(1+r)}, & \text{if } r \leq k, \\ 0, & \text{otherwise.} \end{cases}$$

nDCG@k highlights whether the correct prediction is placed near the very top of the model's candidate list.

These three metrics together provide a comprehensive evaluation: AUC captures global rank discrimination, MRR emphasizes early precision, and nDCG@k measures the quality of the truncated top-$k$ predictions.

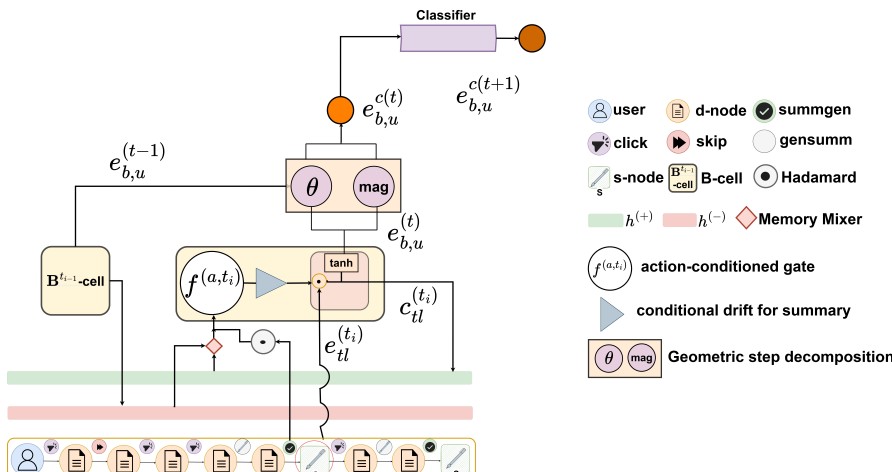

Figure 2: **Walk2Pers-Encoder:** The **b-cell** generates a b-node embedding $e_{b,u}^{(t_i)}$ at timestep $t_i$ using the tail-node embedding $e_{tl}^{(t_i)}$, the action embedding $e_a^{t_i}$, and the history $h^{(t_{i-1})}$ from the previous tail-cell content $c_{tl}^{t_{i-1}}$ (inside the corresponding b-cell); $c_{tl}^{t_{i-1}}$ updates the **dual memory lanes** $(h^+, h^-)$ that persists positive (*click, summarize*) and negative (*skip*) memory resp. to create a **mixed history** $h^{(t_{i-1})}$ which in turn is modulated by a **action-specific gate** $f^{(a,t_i)}$ and a conditional *summGen*-action-triggered **drift** ($\Delta$) before fusing with the tail-node $e_{tl}^{(t_i)}$; the generated $e_{b,u}^{(t_i)}$ then goes through a **geometric-step decomposer** to re-orient the embedding w.r.t continuity vs. novelty.

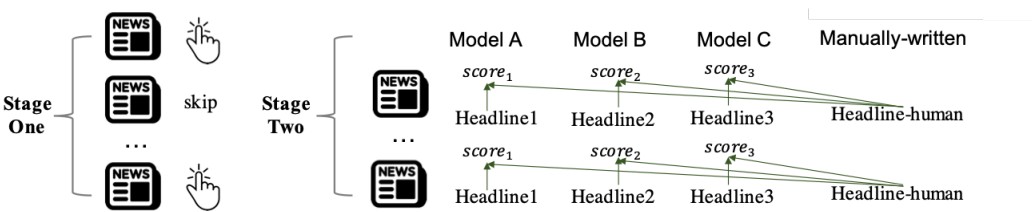

Figure 3: Stages of creation of testing dataset consisting of personalized headlines

## C    DATASETS

### C.1    PENS DATASET

The PENS dataset (Ao et al., 2021) includes 113,762 news articles across 15 topics. Each article contains an ID, title (avg. 10.5 words), body (avg. 549 words), and category, with titles linked to WikiData entities. The dataset also includes user interaction data, such as impressions and click behaviors, combined with news bodies and headlines from the MIND dataset Wu et al. (2020)

**PENS training set.** For training, 500k user-news impressions were sampled from June 13 to July 3, 2019. Each log records user interaction as [uID, tmp, clkNews, uclkNews, clkedHis], where 'clkNews' and 'uclkNews' represent clicked and unclicked news, and 'clkedHis' refers to the user's prior clicked articles, sorted by click time. The training data for Walk2Pers, as discussed in Section 4.1, shows high preference shift. This inherently supports that personalizing UX is strongly dependent on the temporal dynamics of the user. The stats are in the table 6.

**PENS test set.** To create an offline testbed, 103 English-speaking students reviewed 1,000 headlines in stage-1, and then selected 50 articles, and created preferred headlines (i.e., expected gold-reference summaries) for 200 unseen articles in stage-2 (see Figure 3). Each article was reviewed by four participants. Editors checked for factual accuracy, discarding incorrect headlines. The high-quality remaining headlines serve as personalized gold-standard references in the PENS dataset.

Table 6: **MS/CAS PENS Dataset and Interaction Statistics**

| Characteristic | Dimension | Value |
|---|---|---|
| **Article Stats** | | |
| **General Stats** | # Topics | 15 |
| | # Articles | 113,762 |
| | Avg. Title Length | 10.5 words |
| | Avg. Body Length | 549 words |
| **Train Dataset Statistics** | | |
| **Interaction Data** | # User–News Impressions (anon.) | 500,000 |
| | # Users (anon.) | 445,000 |
| | Time Period | June 13–July 3, 2019 |
| | User Interaction Fields | [uID, tmp, clkNews, uclkNews, clkedHis] |
| **Test Dataset Statistics** | | |
| **Participant Stats** | # Participants | 103 |
| | Participant Category | English-speaking college students |
| | # Articles | 3,940 |
| | Browsed Headlines (Click + Skip) | 1,000 per participant |
| | Min. Interested (Click) Headlines | 50 per participant |
| **Gold Reference (Participant-written Headlines)** | Summarized Article Bodies | 200 per participant |
| | Avg. Summaries per Article | 4 |

Table 7: **OpenAI TL;DR (Reddit) Dataset Statistics**

| Characteristic | Dimension | Value |
|---|---|---|
| **Dataset Overview** | | |
| **General Stats** | # Reddit Posts | 123,169 |
| | # Subreddits (Domains) | 29 |
| | Policy-Generated Summaries | 115,579 |
| | Human-Written Summaries | Available |
| **Train + Validation Dataset Statistics** | | |
| **Article Stats** | # Reddit Posts | 21,111 |
| | # Policies | 81 |
| | # Generated Summaries | 107,866 |
| | # Annotators | 76 |
| | # Summary-Pairs Rated | 64,832 |
| Validation Subset Statistics | | |
| **Subset Details** | # Reddit Posts | 1,038 |
| | # Policies | 13 |
| | # Generated Summaries | 7,713 |
| | # Annotators | 32 |
| **Test Dataset (RLHF-Tuned Policies) Statistics** | | |
| **Evaluation Stats** | # Evaluated Policies | 4 |
| | # Evaluated Reddit Posts | 57 (out of 1,038) |
| | Evaluation Method | Indirect Benchmarking |
| **Annotation and Feedback** | | |
| **Feedback Collection** | Rating Scale | 1–7 |
| | Confidence Scale | 1–9 |
| | Avg. Ratings per Annotator | 1,176 |
| | Annotation Format | Summary-Pairs Selection |

## C.2 OPENAI (REDDIT) DATASET

The OpenAI (Reddit) dataset (Völske et al., 2017) comprises 123,169 Reddit posts collected from 29 distinct subreddits. This dataset provides both OpenAI-generated and human-written summaries and is organized into two splits: Comparisons, used for training and validation, and Axis, designated for validation and testing. A curated subset of 1,038 posts was processed by 13 different summarization policies, resulting in the generation of 7,713 summaries. These summaries underwent evaluation by 64 annotators who rated paired summaries based on selection preferences, confidence in their ratings, and dimensions such as accuracy, coherence, coverage, and overall quality. Notably, unlike datasets like PENS, these summaries are not linked to individual annotators or their reading histories, which means they lack elements of personalization and contextual user information. Stats are in Table 7

Table 8: **PersonalSum Dataset Statistics**

| Characteristic | Dimension | Value |
|---|---|---|
| **Dataset Overview** | | |
| **General Stats** | # Articles | 441 |
| | # Annotators | 39 |
| | # Personalized Summaries | 1,099 |
| | Avg. Personalized Summaries per Article | ∼3 |
| **Summary Types** | | |
| **Summary Stats** | GPT-4 Generic Summaries (Post-edited) | 441 |
| | Human-Personalized Summaries | 1,099 |
| | Avg. Summaries per Annotator | ∼28 |
| **Annotation Process** | | |
| **Stages** | Stage-1: Generic Summary Creation | GPT-4 → Human Editing + Highlights |
| | Stage-2: Personalized Summarization | Crowdworkers + Source Highlighting |
| | Stage-3: Quality Filtering | GPT-3.5 Scoring (Coherence, Relevance, Consistency) |
| | Final Dataset | High-quality, aligned summaries with anonymized metadata |
| **Annotation Details** | | |
| **Feedback and Grounding** | Source Sentence Highlighting | Yes |
| | Comprehension Questions | Yes |
| | User Metadata | Anonymized |

Figure 4: **UIG Construction**: Construction of User-Interaction Graph from preference datasets.

## C.3    PERSONALSUM DATASET

The PersonalSum dataset (Zhang et al., 2024) is a crowd-annotated benchmark designed for personalized summarization. It consists of 441 Norwegian news articles across diverse topics and a total of 1,099 human-written summaries contributed by 39 unique annotators. Each article is paired with a GPT-4-generated generic summary (post-edited by students for fluency and factuality), and on average, 3 personalized summaries, written by crowdworkers to reflect their individual preferences and topical interests. Annotators also highlight the source sentences from the article that informed their summaries, adding explicit grounding for each summary segment.

**PersonalSum annotation process.** The dataset was constructed in three stages. In Stage-1, a GPT-4-generated generic summary was created for each article, refined by human editors, and tagged with sentence-level source highlights. In Stage-2, crowdworkers were recruited to write summaries from a personal perspective. Each worker completed comprehension questions and highlighted source text, ensuring fidelity to the original article. In Stage-3, automatic quality checks using GPT-3.5-based scoring (evaluating coherence, relevance, and consistency) filtered poor-quality annotations. Only those scoring above a threshold were retained. The final dataset comprises high-quality user-personalized summaries with source alignments and user metadata anonymized for privacy.

## C.4    UIG CONSTRUCTION

To construct a User-Interaction Graph (UIG) from preference datasets, we distinguish between two types: (i) trajectory-based datasets like PENS Ao et al. (2021), which directly encode user interactions, and (ii) feedback-based datasets like OpenAI-Reddit Völske et al. (2017), which lack user trajectories but contain document nodes (d-nodes), model-generated summaries (s-nodes), and subjective user feedback (e.g., ratings with confidence scores). For PENS-style datasets, we first build a trajectory pool $\mathcal{T}^P$ using click and skip interactions (e.g., *clkNews* and *uclkNews*), but this yields an incomplete user profile due to the absence of personalized s-nodes. To address this, we augment $\mathcal{T}^P$ with s-nodes from the test set by inserting `summarize` and `summGen` edges at the appropriate time

Table 9: **User-Interaction Graph Statistics** for our $\mathcal{T}_{\text{train}}^{\text{PENS-D}}$, $\mathcal{T}_{\text{train}}^{\text{OAI}}$ and $\mathcal{T}_{\text{train}}^{\text{PS-EN}}$.

| Characteristic | $\mathcal{T}_{\text{train}}^{\text{PENS-D}}$ | $\mathcal{T}_{\text{train}}^{\text{OAI}}$ | $\mathcal{T}_{\text{train}}^{\text{PS-EN}}$ |
|---|---|---|---|
| # u-nodes (trajectories) | 55,000 | 18,000 | 700 |
| # d-nodes per trajectory | 134 | 39 | 17 |
| # s-nodes per trajectory | 15 | 12 | 4 |
| Average trajectory length | 143 | 47 | 23 |
| # Max. trajectory length | 200 | 50 | 32 |
| # Min. trajectory length | 5 | 25 | 7 |
| Rate of Topic Shift | 0.77 | 0.48 | 0.41 |

steps, resulting in a derived pool $\mathcal{T}^{\text{P-D}}$. For OpenAI-style datasets, we infer preference sequences by classifying $d$-nodes as clicked if any associated system-generated summary received a rating above a threshold (i.e., greater than equals to 6 out of 9), then selecting the highest-rated summary as a surrogate $s$-node. The UIG construction algorithm is described in Algorithm 1.

---

**Algorithm 1** UIG Construction
---
0: **function** CONSTRUCT_UIG(train, test_data, type)
1: Initialize $\mathcal{T}_{\text{PENS}}, \mathcal{T}_{\text{OAI}} \leftarrow \emptyset$
2: **for** each user $u$ in train_data **do**
3:    Initialize $\tau^u \leftarrow \emptyset$
4:    **for** each interaction in $u$'s data **do**
5:       **if** type = PENS **then**
6:          Map to d-node with a *click/skip* edge based on interaction
7:       **else**
8:          Map to d-node with *click/skip* edge based on rating
9:          **if** rating confidence = max **then**
10:             Map to d-node with *gensum*, and s-node with *sumgen* edge
11:          **end if**
12:       **end if**
13:    Append mapped d-node to $\tau^u$
14:    **end for**
15:    Add $\tau^u$ to $\mathcal{T}_{\text{PENS}}$ or $\mathcal{T}_{\text{OAI}}$ based on type
16: **end for**
17: **if** type = PENS **then**
18:    **for** $\tau^u$ in $\mathcal{T}_{\text{PENS}}$ **do**
19:       Retrieve and insert s-nodes from test_data using *genSumm/sumgen* edges
20:    **end for**
21:    **return** $\mathcal{T}^{\text{PENS-D}} \leftarrow \mathcal{T}_{\text{PENS}}$
22: **else**
23:    **return** $\mathcal{T}_{\text{OAI}}$
24: **end if**
24: **end function**=0

---

## D THEORETICAL ANALYSIS: SWH STRICTLY GENERALIZES MDH

We provide formal derivations showing that the Structured Walk Hypothesis (SWH) strictly generalizes the Markovian Drift Hypothesis (MDH).

**Lemma D.1** (Reduction: SWH recovers MDH). *For any MDH update of the form*

$$e_{b,u}^{(t+1)} = f(e_{b,u}^{(t)}, a^{(t)}, q) + \epsilon^{(t)}, \quad \epsilon^{(t)} \sim \mathcal{N}(0, \Sigma(a^{(t)})),$$

*there exists a parameter setting of SWH (Eq. 3) that reproduces it.*

*Proof.* Set $\Psi \equiv 0$ and $\delta \equiv 0$. The increment under MDH is $\Delta e^{(t)} = f(e_{b,u}^{(t)}, a^{(t)}, q) - e_{b,u}^{(t)}$. We can select $u^{(t)} = \Delta e^{(t)}/\|\Delta e^{(t)}\|$, any $o^{(t)} \perp u^{(t)}$, and set $\text{mag}(a^{(t)})$ to $\text{mag}(a^{(t)}) = \|\Delta e^{(t)}\|$,

$\theta(a^{(t)}) = 0$. Then $\Phi = \Delta e^{(t)}$, and Eq. 3 collapses to the MDH update. Noise $\epsilon^{(t)}$ is matched by sampling mag or adding a Gaussian head tied to $a^{(t)}$. $\qquad\square$

**Corollary D.2** (MDH variants as special cases). *The following well-known models are recovered as degenerate cases of SWH:*

- *Pure Random Walk (PRW):* $\mathrm{mag} = 0$, $\Psi = 0$, $\delta = 0$.

- *Action-aware Random Walk (ARW): Same as PRW, but with action-conditioned noise* $\epsilon^{(t)} \sim \mathcal{N}(0, \Sigma(a^{(t)}))$.

- *Random Walk with Restart (RWR): Tong et al. (2006) proposed this popular diffusion model where, at each step, the walk either (i) follows the graph transition with probability $(1 - \alpha)$ or (ii) "restarts" to a fixed seed distribution $q$ with probability $\alpha$. This ensures the walk remains centered around the most recent interaction and does not drift arbitrarily far. In our SWH framework, this corresponds to choosing* $\Phi = \alpha (q - e_{b,u}^{(t)})$ *with* $\theta = 0$ *and* $\alpha \in (0, 1)$.

**Lemma D.3** (Short-memory encoders are contained). *Any continuous short-memory update* $\tilde{f}(e_{b,u}^{(t)}, a^{(t)})$ *(e.g., GRU, attention block) is realizable by SWH with* $\Psi = \delta = 0$, *since* $\Phi = \tilde{f}(e_{b,u}^{(t)}, a^{(t)}) - e_{b,u}^{(t)}$.

**Theorem D.4** (Strict Generalization). *SWH strictly generalizes MDH:*

1. *Every MDH update is realizable by SWH (Lemmas D.1–D.2).*

2. *There exist SWH updates that no MDH update can reproduce.*

*Proof.* (i) follows from the lemmas. (ii) Consider two histories $\mathcal{H}_1, \mathcal{H}_2$ with identical $(e_{b,u}^{(t)}, a^{(t)})$ but distinct memories $(h_t^+, h_t^-)$. Eq. 3 yields increments $\Phi + \Psi(h_t^+, h_t^-) + \delta$ and $\Phi + \Psi(h_t'^+, h_t'^-) + \delta$, which differ. Since MDH is first-order Markov in $(e_{b,u}^{(t)}, a^{(t)}, q)$, it cannot separate $\mathcal{H}_1$ from $\mathcal{H}_2$. Thus SWH has strictly greater expressivity. $\qquad\square$

**Corollary D.5** (Non-Markovianity from dual memory). *If $\Psi$ aggregates asymmetric signals over multiple timesteps, the resulting process is non-Markovian with respect to the pair $(e_{b,u}^{(t)}, a^{(t)})$, and hence lies outside the scope of MDH.*

*Proof Sketch.* Under MDH (Eq. 1), the next state depends only on the *current* preference embedding $e_{b,u}^{(t)}$ and action $a^{(t)}$. Thus two different histories $\mathcal{H}_1$ and $\mathcal{H}_2$ that lead to the same $(e_{b,u}^{(t)}, a^{(t)})$ are indistinguishable. In contrast, SWH retains separate reinforcement and suppression traces via dual memories $(h_t^+, h_t^-)$. If $\mathcal{H}_1$ contains repeated clicks on climate policy and $\mathcal{H}_2$ contains repeated skips of celebrity news, both may yield the same $e_{b,u}^{(t)}$, although $(h_t^+, h_t^-)$ differ. Consequently, $\Psi(h_t^+, h_t^-)$ produces distinct updates in Eq. 3. Therefore the transition distribution $P(e_{b,u}^{(t+1)} \mid e_{b,u}^{(t)}, a^{(t)}, \mathcal{H}_i)$ depends on the full history $\mathcal{H}_i$, not just the present state, violating the Markov property. $\qquad\square$

**Remark.** This shows that even when the visible preference embedding $e_{b,u}^{(t)}$ is the same, Walk2Pers can differentiate users with distinct long-term interaction traces. This is impossible under MDH, where all memory beyond the latest step is collapsed into $q$. In other words, SWH reduces to MDH when memory and drift are disabled, but enables trajectories to diverge even when current states coincide. This matches the empirical evidence that user preferences depend on long-horizon, action-specific dynamics (see Table 3).

# E  BASELINES

## E.1  BASELINES FOR RQ-1/2: TESTING MDH VS. SWH COMPONENTS

To evaluate whether the Markovian Drift Hypothesis (MDH) suffices or if the Structured Walk Hypothesis (SWH) is necessary, we compare Walk2Pers against a suite of encoder variants and established news recommendation encoders:

Table 10: **Encoder Models are Markovian Drifters**: `Walk2Pers` uniquely integrates all aspects of SWH; *history capped by context length; no persistent memory across examples.*

| Models | History | Memory | Action Conditioning | Explicit Step | Long-term Persistent Memory | Interpretability |
|---|---|---|---|---|---|---|
| Oracles (BigBird-Pegasus, SimCLS, T5-base) | ✗ | ✗ | ✗ (gold injected as cues) | ✗ | ✗ | ✗ |
| LLMs (Zephyr, Mistral, LLaMA-2, DeepSeek) | ✓ (injected in prompt) | ✗ | ✗* | ✗ | ✗ | ✗ |
| PRW | ✓ | ✗ | ✗ | ✗ | ✗ | ✗ |
| SMD (Short-Memory Drift) **(ours)** | ✓ | ✗ | ✗ | ✗ | ✗ | ✗ |
| ARW | ✓ | ✗ | ✓ | ✗ | ✗ | ✗ |
| RWR / D-RDW | ✓ | ✗ | ✓ | ✗ | ✗ | ✗ |
| AGD (Action-Gated Drift) **(ours)** | ✓ | ✗ | ✓ | ✗ | ✗ | ✗ |
| NAML | ✓ | ✓(short) | ✓ | ✗ | ✗ | ✗ |
| NRMS | ✓ | ✓(short) | ✓ | ✗ | ✗ | ✗ |
| EBNR | ✓ | ✓(short) | ✓ | ✗ | ✗ | ✗ |
| GTP (w/ TrRMIo) | ✓ | ✓(injected short) | ✓ | ✗ | ✗ | ✗ |
| SP (Signature-Phrase) | ✓ | ✓(phrases) | ✓ | ✗ | ✗ | ✗ |
| **Walk2Pers (ours)** | ✓ | ✓(dual) | ✓ | ✓ | ✓ | ✓ |

**Short-Memory Gate (SMD).** This is the minimal baseline consistent with the Markovian Drift Hypothesis. At timestep $t_i$, the update depends only on the immediate tail state and current action embedding via a convex gate:

$$c_{\text{tl}}^{(t_i)} = \tanh\big(f_{\text{SMD}}^{(a,t_i)}\big) \, \odot \, e_{\text{tl}}^{(t_i)}, \qquad e_{b,u}^{(t_i)} = \tanh\big(W_b \, c_{\text{tl}}^{(t_i)}\big),$$

where the gate is defined as: $f_{\text{SMD}}^{(a,t_i)} = \eta\big(W_a \cdot e_a^{(t_i)}\big) + (1-\eta)\big(W_h \cdot c_{\text{tl}}^{(t_{i-1})}\big)$. Here, $\eta \in (0,1)$ is a learnable scalar which balances the contribution of the action embedding $e_a^{(t_i)}$ and the prior tail state $c_{\text{tl}}^{(t_{i-1})}$. Because this update overwrites history at each step, it carries no mechanism for reinforcement, suppression, or geometric decomposition, and thus strictly follows the MDH assumption.

**Action-Specific Gates (AGD).** Instead of a generic convex combination, each action has its own parameterization of $f(\cdot)$. We design $f^{(a,t_i)}$ separately for `click`, `skip`, `summarize`, and `summGen`, so that each action boosts, mutes, or rebalances user interests. A *click* encodes positive reinforcement, implemented as a history-aware gain:

$$f^{(clk,t_i)} = \big(\mathbf{W}_{\text{clk}} \cdot \mathbf{e}_{clk}^{(t_i)}\big) \odot \mathbf{c}_{tl}^{(t_{i-1})}, \quad \mathbf{c}_{tl}^{(t_i)} = \tanh(f^{(clk,t_i)}) \odot \mathbf{e}_{tl}^{(t_i)}. \tag{17}$$

If Alice clicks on "*Concert in New York*", the gain reinforces music-related features in her history. In contrast, *skip* reflects either disinterest in the current document or *pull* toward an alternative. In terms of the timesteps of the trajectory $\tau_u$, this would be a look-ahead to timestep $t_{i+1}$ where $\mathbf{c}_{tl}^{(t_i)}$ is compared with $\mathbf{e}_{tl}^{(t_{i+1})}$ via a dot product $\langle \cdot \rangle$. We model this alternative attraction towards a more preferred (in contrast to disinterest) alternative as a deviation $\eta_{attr}^{(t_i)}$:

$$f^{(skp,t_i)} = \tanh\Big(\mathbf{W}_{\text{skp}}\mathbf{e}_{skp}^{(t_i)} + \mathbf{g}_{move}^{(t_i)}\Big) \odot \mathbf{e}_{tl}^{(t_i)}; \text{where: } \mathbf{g}_{move}^{(t_i)} = \lambda_1^{(t_i)}\mathbf{e}_{tl}^{(t_i)} + \mathbf{W}_{\text{pull}} \cdot \lambda_2^{(t_i)}\eta_{attr}^{(t_i)};$$
$$\eta_{attr}^{(t_i)} = \max\big(\langle \mathbf{c}_{tl}^{(t_{i-1})}, \mathbf{e}_{tl}^{(t_{i+1})}\rangle, \, 1 - \langle \mathbf{c}_{tl}^{(t_{i-1})}, \mathbf{e}_{tl}^{(t_i)}\rangle\big); \mathbf{c}_{tl}^{(t_i)} = \tanh(f^{(skp,t_i)}) \odot \mathbf{e}_{tl}^{(t_i)} \tag{18}$$

Here, $\mathbf{g}_{move}^{(t_i)}$ non-linearly distorts the trajectory. $\lambda_1, \lambda_2$ are learnable constants. To illustrate, Alice may choose to skip a *Election Policies* article (due to disinterest) or leave, say "*Music During Cooking*", for a more preferred "*Latest Concerts by Pink Panther*" (i.e., the future pull).

A summary request, i.e. `summarize` indicates focused intent anchored in the title of the document:

$$f^{(summ,t_i)} = (\mathbf{W}_{summ} \cdot \mathbf{e}_{summ}^{(t_i)}) \odot \mathbf{c}_{tl}^{(t_{i-1})}, \quad \mathbf{c}_{tl}^{(t_i)} = \tanh(f^{(summ,t_i)}) \odot \mathbf{e}_{\mathbf{title}(tl)}^{(t_i)}. \tag{19}$$

Finally, generated summary `summGen` evaluates whether the summary is concise in terms of expectations and faithful to the source content:

$$f^{(summGen,t_i)} = -\gamma \log\Big( \exp(\mathbf{g}_{\text{summGen}} \odot \mathbf{e}_{tl}^{(t_i)}) + \exp((1-\mathbf{g}_{\text{summGen}}) \odot \mathbf{e}_{tl}^{(t_{i-1})}) \Big),$$
$$\mathbf{c}_{tl}^{(t_i)} = \tanh(f^{(summGen,t_i)}) \odot \mathbf{c}_{tl}^{(t_{i-1})}, \quad \mathbf{g}_{\text{summGen}} = \sigma\big(\mathbf{W}_{\text{min}}(\mathbf{W}_{\text{summGen}} \cdot \mathbf{e}_{\text{summGen}}^{(t_i)})\big). \tag{20}$$

The gate $\mathbf{g}_{\text{summGen}}$ balances coverage of the source along with history alignment, with a learnable $\gamma$ tuning the trade-off between them. We generalize the gates as: $f^{(a,t_i)} = \text{AGD}(\mathbf{e}_a, t_i) \odot \mathbf{e}_{\text{history}}$, where AGD determines which function to trigger based on the action $a$ and $\mathbf{e}_{\text{history}}$ is the embedding of the previous state history.

### E.2 Baselines for RQ-3 (Personalized Summarization Performance)

For downstream evaluation on personalized summarization, we compare against three categories of strong baselines:

#### E.2.1 Baseline Personalized Models

**PENS-NRMS Injection-Type 1.** The PENS framework (Ao et al., 2021) generates personalized summaries by incorporating user embeddings along with the input news article. For this variant, user embeddings are derived using NRMS (Neural News Recommendation with Multi-Head Self-Attention) (Wu et al., 2019b), which includes a multi-head self-attention based news encoder to represent news titles, and a user encoder that captures browsing behavior through multi-head self-attention over clicked articles. Additive attention mechanisms are employed to highlight important words and articles. In Injection-Type 1, the NRMS user embedding is injected by initializing the decoder's hidden state, thereby directly influencing the summary generation process from the start.

**PENS-NRMS Injection-Type 2.** This variant also uses NRMS for user embedding, but personalization is introduced differently. Instead of initializing the decoder, the user embedding is injected into the attention mechanism of the PENS model. This modulates the attention weights over the news body, enabling the model to focus on content aligned with the user's preferences.

**PENS-NAML Injection-Type 1.** NAML (Neural News Recommendation with Attentive Multi-View Learning) (Wu et al., 2019a) generates news representations by attending over multiple views, including titles, bodies, and topic categories. The user encoder learns from interacted news and selects the most informative content for personalization. The resulting user embedding is integrated into the PENS decoder using Injection-Type 1, i.e., by initializing the decoder's hidden state.

**PENS-EBNR Injection-Type 1.** EBNR (Embedding-based News Recommendation) (Okura et al., 2017) models user preferences using an RNN over browsing histories to produce user embeddings. These embeddings are injected into the PENS model via Injection-Type 1 by initializing the decoder, thereby influencing the initial decoding steps with user-specific information.

**PENS-EBNR Injection-Type 2.** This configuration uses the same user encoder from EBNR but applies Injection-Type 2. Here, the user embedding is incorporated into the decoder's attention layers, allowing the model to personalize attention distributions over the news body during decoding.

**General Then Personal (GTP).** General Then Personal (GTP) (Song et al., 2023) is a two-stage framework for personalized headline generation. In stage-1, a Transformer-based encoder–decoder model is pre-trained on large-scale news article–headline pairs to learn robust, content-focused headline generation without personalization. In stage-2, a separate "headline customizer" refines the general headline by incorporating user-specific preferences, which are encoded as a control code by the user encoder TrRMIo. To bridge the gap between general generation and personalized refinement, GTP introduces two mechanisms: (i) **Information Self-Boosting (ISB)**, which reintroduces relevant content details from the article to prevent information loss during customization; and (ii) **Masked User Modeling (MUM)**, which randomly masks parts of the user embedding during training and reconstructs them, reducing the model's over-reliance on its general parameters.

**Signature Phrase.** Another line of personalization focuses on condensing a user's reading history into a collection of *signature phrases* (Cai et al., 2023). These phrases, derived through contrastive learning over news articles without annotated data, act as dynamic user profiles that adapt as interests evolve. Such phrases need not appear verbatim in the user's history but instead encode higher-level signals. Using these phrases, the model learns to generate personalized headlines that connect new articles with the user's inferred interests, yielding outputs that are engaging, relevant, and grounded in article content rather than drifting toward clickbait.

These encoders serve as competitive MDH-aligned baselines, since they reduce trajectories into compressed short-term embeddings with no explicit long-term reinforcement or novelty modeling.

### E.2.2 BASELINE LLMS

**Zephyr 7B** $\beta$**.** Zephyr(Tunstall et al., 2023) is a 7 billion parameter transformer model fine tuned from Mistral 7 billion using Direct Preference Optimization on publicly available and synthetic data. It removes traditional alignment constraints to improve raw performance and achieves strong results on benchmarks such as MT Bench where it reports a score of 7.34 in comparison with 6.86 for LLaMA 2-70B Chat. Zephyr is optimized for helpful dialogue and is openly available under an MIT license. It focuses on efficiency and high quality responses without relying on reinforcement learning from human feedback. The model supports an input context length of 32K tokens.

**Mistral 7B.** Mistral Instruct(Jiang et al., 2023) is a dense transformer model that uses grouped query attention and sliding window attention to efficiently scale with long context inputs. It is pretrained on approximately two trillion tokens and provides strong performance across natural language and code generation benchmarks, surpassing models such as LLaMA 2-13B in many evaluations. Mistral Instruct is fully open source under the Apache 2.0 license and includes an instruction tuned variant that is widely adopted for fine tuning and deployment. The model supports an input context length of 32K tokens.

**LLaMA 2 13B.** LLaMA two(Touvron et al., 2023) 13 billion by Meta is an autoregressive transformer trained on two trillion tokens of public data with a context length of 4096 tokens. It supports chat interaction through instruction tuning and reinforcement learning from human feedback. Although originally state of the art among open models, it is surpassed in many tasks by newer architectures such as Mistral-7 billion. LLaMA-2 remains an influential and widely used foundation model with extensive documentation and open access under the Meta license.

**DeepSeek R1 14B.** DeepSeek R1(DeepSeek-AI et al., 2025) is a 14.8 billion parameter model distilled from Qwen-2.5 14 billion and is designed for strong performance on mathematical reasoning, coding, and multi step logical tasks. It is fine tuned on eight hundred thousand examples generated by a larger DeepSeek R1 model and is released under an MIT license. Despite its moderate size, it rivals substantially larger models on benchmarks such as AIME and MATH while remaining efficient for customization and deployment. The model supports an input context length of 128K tokens.

**Gemini 2.5 Flash.** Gemini-2.5 Flash is a compact and highly optimized member of the Gemini family designed for fast inference, large scale retrieval augmented generation, and multimodal interaction. It provides high throughput generation with reduced latency and is suitable for production applications that require consistent responsiveness. Gemini 2.5 Flash incorporates the same unified multimodal architecture used across the Gemini series and has been trained on a large mixture of text, code, and image data. The model supports an input context length of 1 million tokens, making it particularly effective for long document synthesis and extended conversational sessions.

**Qwen 3 235B.** Qwen-2 235 billion is a large scale frontier transformer model developed as part of the Qwen three family. It is trained on extensive multilingual and multimodal datasets with a focus on reasoning, tool use, and high fidelity generation. The model is designed for advanced analytical tasks, multi hop reasoning, and instruction following at very high capability levels. Qwen-3 serves as a foundation for many distilled and specialized variants and is openly available for research and commercial use under a permissive license. The model supports an input context length of 256K, allowing it to operate effectively on extremely long sequences in both textual and mixed modality settings.

### E.2.3 BASELINE GENERIC SUMMARIZERS

**BigBirdPegasus.** BigbirdPegasus, proposed by (Zaheer et al., 2020) is an extension of Transformer based models designed specifically for processing longer sequences. It utilizes sparse attention, global attention, and random attention mechanisms to approximate full attention. This enables BigBird to handle longer contexts more efficiently and, therefore, can be suitable for summarization.

**SimCLS.** A Simple Framework for Contrastive Learning of Abstractive Summarization (Liu & Liu, 2021) uses a two-stage training procedure. In the first stage, a Seq2Seq model (Lewis et al.,

2020) is trained to generate candidate summaries with MLE loss. Next, the evaluation model, initiated with RoBERTa is trained to rank the generated candidates with contrastive learning.

**T5.** (Text-To-Text Transfer Transformer) is based on the Transformer-based Encoder-Decoder architecture that operates on the principle of the unified text-to-text task for any NLP problem, including summarization. Some recent analyses on the performance of T5 on summarization tasks can be found in (Raffel et al., 2020; Tawmo et al., 2022; Ramesh et al., 2022; Etemad et al., 2021).

## F  LICENSE AND USAGE STATEMENT

In this work, we utilize the following pre-trained large language models (LLMs): DeepSeek-R1 14B (MIT License), Mistral-7B-Instruct (Apache 2.0), LLaMA2-13B (Llama 2 Community License), and Zephyr 7B ($\beta$) (MIT License). All models are used according to their respective licenses and terms provided by their original creators. Proper attribution is given to each model's developers as cited in our references. We also use the following datasets:

- **MS/CAS PENS dataset:** We comply with the dataset's terms of use, which is derived from the Microsoft Research License (`https://github.com/msnews/MIND/blob/master/MSR%20License_Data.pdf`).
- **OpenAI Reddit dataset:** We comply with the MIT License specifications as set by OpenAI (`https://github.com/openai/summarize-from-feedback/blob/master/LICENSE`)

We have ensured that all datasets and models are used responsibly, respecting privacy, consent, and ethical guidelines. When applicable, data is anonymized and handled according to the ethical standards set forth by NeurIPS.

## G  WALK2PERS ARCHITECTURAL DETAILS

### G.1  WALK2PERS-ENCODER ARCHITECTURAL DETAILS

**Magnitude and Orientation Computation.** To capture the *direction* and *magnitude* of movement in a complex embedding-based manifold, we proposed a geometrical walk of behaviors 3.3.1. This walk conceptualizes preference evolution as a sequence of *directed steps* in semantic space where each behavior is obtained by rotating and scaling the previous embedding to get the geometrically contextualized $\mathbf{e_c}_{b_{u_j}}^{(t_i)}$. We compute the **orientation degree of the update** as follows:

$$\theta^{(t_i)} = \pi \cdot \tanh\Big(\mathbf{W}_\theta \, \sigma\big(\mathbf{W}_{angle} \, \mathbf{e}_{b,u}^{(t_{i-1})}\big)\Big); \quad o^{(t_i)} = \frac{v^{(t_i)} - \langle v^{(t_i)}, u^{(t_{i-1})}\rangle \, u^{(t_{i-1})}}{\max\big(\|v^{(t_i)} - \langle v^{(t_i)}, u^{(t_{i-1})}\rangle \, u^{(t_{i-1})}\|_2, \ \varepsilon\big)},$$

$$v^{(t_i)} = \frac{\mathbf{e}_{b,u}^{(t_i)} - \mathbf{e}_{b,u}^{(t_{i-1})}}{\max\big(\|\mathbf{e}_{b,u}^{(t_i)} - \mathbf{e}_{b_{u_j}}^{(t_{i-1})}\|_2, \ \varepsilon\big)}; \quad u^{(t_{i-1})} = \frac{\mathbf{e}_{b,u}^{(t_i)}}{\max\big(\|\mathbf{e}_{b,u}^{(t_i)}\|_2, \ \varepsilon\big)}.$$

$$(21)$$

The angle $\theta^{(t_i)} \in (-\pi, \pi)$ serves as a *directional drift controller*, deciding whether the trajectory should continue smoothly (small $\theta^{(t_i)}$) or shift sharply (large $\theta^{(t_i)}$). The vector $u^{(t_{i-1})}$ represents the *forward direction* inherited from past context (carrying forward the momentum of current flow of history), while $o^{(t_i)}$ denotes the *orthogonal novelty axis* derived from the raw transition direction $v^{(t_i)}$. For example, if Alice has consistently read *Climate Science Articles*, $u^{(t_{i-1})}$ encodes this momentum, while a sudden click on a *Policy Debate* yields a transition vector with a strong orthogonal component $o^{(t_i)}$; a small $\theta^{(t_i)}$ implies smooth thematic extension (science into policy), whereas a large $\theta^{(t_i)}$ reflects a sharp diffusion toward politics.

To determine how far $\mathbf{e}_{b,u}^{(t_i)}$ travels along the chosen direction, the **magnitude of movement** is regulated as:

$$mag^{(t_i)} = \text{Softplus}\Big(\mathbf{W}_m \cdot \mathbf{W}_h \cdot \mathbf{e}_{b,u}^{(t_{i-1})}\Big),$$

$$(22)$$

where $mag^{(t_i)}$ acts like the *distance* of the walk, dictating whether the model advances cautiously or moves decisively.

$$\mathbf{e_{c}}_{b_{u_j}}^{(t_i)} = \mathbf{e}_{b,u}^{(t_i)} + mag^{(t_i)}\Big( \cos\theta^{(t_i)} \cdot u^{(t_{i-1})} + \sin\theta^{(t_i)}\, o^{(t_i)} \Big). \tag{23}$$

This ensures that preference evolution is represented as a directed step, combining alignment with historical momentum and deviation toward orthogonal novelty. For Alice, this means her reading trajectory can smoothly extend within climate science when $\theta^{(t_i)}$ is small and $mag^{(t_i)}$ is low (steady interest), or make a decisive turn into politics when $\theta^{(t_i)}$ is large and $mag^{(t_i)}$ accelerates the shift (sharp interest change).

## G.2   WALK2PERS-ENCODER AS AN INSTANTIATION OF SWH

We now formalize how the `Walk2Pers`-encoder concretely realizes the SWH update (Eq. 3). For reference, recall the SWH rule:

$$
\begin{aligned}
e_{b,u}^{(t+1)} \;=\; e_{b,u}^{(t)} + \underbrace{\mathrm{mag}(a^{(t)})\big( \cos\theta(a^{(t)})\, u^{(t)} + \sin\theta(a^{(t)})\, o^{(t)} \big)}_{\Phi:\text{ geometric step}} \\
+ \underbrace{\Psi(h_t^+, h_t^-)}_{\text{dual memory}} + \underbrace{\delta \cdot \mathbf{1}[a^{(t)} = \texttt{summGen}]}_{\text{drift}}.
\end{aligned}
\tag{3}
$$

**Theorem G.1** (`Walk2Pers`an instantiation of SWH). *With the definitions of Sec. 3.3.1, the* `Walk2Pers`*-encoder update is exactly of the form Eq. 3.*

*Detailed Sketch.* We match each component of Eq. 3 to the `Walk2Pers`.

**(A) Geometric step.** The encoder produces two scalar heads $(\hat{m}_t, \hat{\theta}_t)$ for each action $a^{(t)}$:

$$\mathrm{mag}(a^{(t)}) = \texttt{softplus}(\hat{m}_t) \quad \geq 0,$$
$$\theta(a^{(t)}) = \pi \cdot \sigma(\hat{\theta}_t) \quad \in (0,\pi).$$

These control the *length* and *orientation* of the step. Next, `Walk2Pers` maintains two orthogonal unit axes: - $u^{(t)}$: the "momentum" direction, aligned with recent preference drift. - $o^{(t)}$: an orthogonal "novelty" axis obtained via Gram–Schmidt. Together, these yield the geometric increment

$$\Phi_t \;=\; \mathrm{mag}(a^{(t)})\big( \cos\theta(a^{(t)}) u^{(t)} + \sin\theta(a^{(t)}) o^{(t)} \big).$$

*Intuition:* This term says each action either pushes the state forward in a continuity-preserving direction (small $\theta$) or rotates into a novel axis (large $\theta$), with strength controlled by $\mathrm{mag}(a^{(t)})$.

**(B) Dual memory.** `Walk2Pers` keeps two asymmetric accumulators:

$$h_t^+ = h_{t-1}^+ + m^{(t)} \odot c^{(t)},$$
$$h_t^- = h_{t-1}^- \odot (1 - m^{(t)}) + c^{(t)}.$$

Here $m^{(t)} = \mathrm{SoftMax}(W_h h^{(t-1)} + W_c c^{(t)})$ is an action gate, and $c^{(t)}$ is the content embedding. Thus: - $h^+$ *reinforces* positively gated interactions (e.g. clicks), accumulating them additively. - $h^-$ *suppresses* negatively gated interactions (e.g. skips), attenuating old signals via $(1 - m^{(t)})$.

The two lanes are blended with a learnable weight $\omega \in (0,1)$:

$$h^{(t)} = \omega h_t^+ + (1-\omega) h_t^-.$$

Finally, a linear projection produces the memory contribution:

$$\Psi(h_t^+, h_t^-) = W_\Psi [h_t^+; h_t^-].$$

*Intuition:* Unlike MDH, which forgets everything beyond the last step, this mechanism lets `Walk2Pers` reinforce long-term positive signals while still allowing suppression of repeated disinterest. Thus $\Psi$ injects non-Markovian, history-dependent bias into the walk.

**(C) Drift.** For the `summGen` action, `Walk2Pers` triggers an extra *summary drift* term. A drift vector $\delta = W_\delta \phi_\delta(x_t)$ is generated and added only when $a^{(t)} = \texttt{summGen}$:

$$\Delta^{(t)} = \delta \cdot \mathbf{1}[a^{(t)} = \texttt{summGen}].$$

*Intuition:* This nudges the preference state toward more "condensed" representations when the user explicitly asks for a summary, acknowledging that summarization requests are qualitatively different from passive clicks or skips.

Putting together (A)–(C), the `Walk2Pers` update rule is

$$e_{b,u}^{(t+1)} = e_{b,u}^{(t)} + \Phi_t + \Psi(h_t^+, h_t^-) + \Delta^{(t)},$$

which is identical to Eq. 3. Hence `Walk2Pers` a concrete instantiation of the Structured Walk Hypothesis. $\qquad\square$

**Remark** By parameter restriction: setting $\Psi \equiv 0$ and $\delta \equiv 0$ collapses `Walk2Pers` to a pure geometric update. Further restrictions recover PRW, ARW, and RWR as shown in Cor. D.2.

### G.3 WALK2PERS-DECODER DETAILS

**Latent Summary Contextualization** The query b-node embedding $\mathbf{e}_{b_q,u}^{(t_{l+1})}$ predicted by the encoder represents a geometrical alignment infused entanglement of behavior duplet $\langle genSumm, s_q\rangle$. The $s_q$ represents the latent s-node embedding of the query document $d_q$. Since $\mathbf{e}_{b_q,u}^{(t_{l+1})}$ has infused the learned action embedding, it becomes difficult for a decoder to feed on it and generate a personalized summary. To address this, we extract the latent s-node embedding from $\mathbf{e}_{b_q,u}^{(t_{l+1})}$.

**Latent s-Node Contextualizer.** Although latent s-node embedding $\hat{\mathbf{e}}_{s_q,u}^{(t_{l+1})}$ represents the user's personalized summary intention, it lacks the contextualization of the query document $d_q$. The base **decoder** contextualizes $d_q$ with $\hat{\mathbf{e}}_{s_q,u}^{(t_{l+1})}$ using cross-attention, and feeds to a summarizer decoder. The query document $\mathbf{e}_{d_q}$ serves as the query, and $\hat{\mathbf{e}}_{s_q,u}^{(t_{l+1})}$ acts as both key and value, resulting summary contextualized query embedding as:

$$\mathbf{e}_{d_q}^c = \text{SoftMax}\left(\frac{(W_q \cdot \mathbf{e}_{d_q})^\top (W_k \cdot \hat{\mathbf{e}}_{s_q,u}^{(t_{l+1})})}{\sqrt{d}}\right) \cdot (W_v \cdot \hat{\mathbf{e}}_{s_q,u}^{(t_{l+1})}) \tag{24}$$

This contextualizes the document to be summarized with the latent summary intention of the user, but the query document $\mathbf{e}_{d_q}$ lacks explicit user-preference representation.

**User-based d-Node Encoding.** To incorporate explicit user preferences into the document representation, we enrich the query document embedding with the user's interaction history. Specifically, the encoder of the summarizer model produces the base document embedding $\mathbf{e}_{d_q}$. In parallel, the final user history vector $\mathbf{e}_{b_q,u}^{(t_l)}$ from the `Walk2Pers` encoder is applied as a gating signal to modulate $\mathbf{e}_{d_q}$, yielding the user-aware document embedding:

$$\mathbf{e}_{d_q,u} = \sigma\left(W_g \cdot \mathbf{e}_{b_q,u}^{(t_l)}\right) \odot \mathbf{e}_{d_q}, \tag{25}$$

The gating makes the query document align with the user's own preference, while irrelevant aspects are suppressed. This ensures the document is passed through the lens of Alice for generating the expected summary for her, and the same document passes through the lens of Bob for generating his expected summary. $\mathbf{e}_{d_q,u}$ encodes both the semantic content of the document and the personalized preference profile of the user, enabling more faithful contextualization in subsequent cross-attention. The latent s-node then contextualizes $\mathbf{e}_{d_q,u}$ to produce $\mathbf{e}_{d_q,u}^c$, as discussed in Section 3.3.2.

**Personalized Summarization.** The `Walk2Pers` decoder generates a personalized summary by feeding on contextualized query document embedding $\mathbf{e}_{d_q,u}^c$. We use the T5-base (Raffel et al., 2020) decoder for the summarization.

**Decoder Training.** The decoder training objective ($\mathcal{L}_{\text{dec}}$) is a linear combination of two loss functions, *Generation Loss* ($\mathcal{L}_{\text{gen}}$) and the earlier encoder loss ($\mathcal{L}_{\text{enc}}$; see Section 3.3.2), as $\mathcal{L}_{\text{dec}} = \beta \cdot \mathcal{L}_{gen} + (1 - \beta) \cdot \mathcal{L}_{\text{enc}}$. Here, $\mathcal{L}_{\text{gen}}$ is the cross-entropy loss between predicted tokens $\hat{y}$ and ground-truth $y^*$ under teacher forcing with the T5 decoder. Optimizing $\mathcal{L}_{\text{gen}}$ updates the cross-attention layers and language-modeling head of T5 decoder, contextualizer weights $W_k$, $W_q$, $W_v$ and inverse-mapping weights $W_{\text{summ}}^+$, $W_c^+$. Fine-tuning the cross-attention layers ensures that the decoder learns how to properly fuse the contextualized document embedding $\mathbf{e}_c^{(d_q, u_j)}$ with the latent s-node embedding, thereby injecting user-specific preference signals into the decoding process, and fine-tuning the language modeling head adapts the token generation distribution to reflect this personalized conditioning, improving lexical and stylistic alignment with the user's history. This ensures accurate latent s-node reconstruction $\hat{\mathbf{e}}_{s_q, u}^{(t_{l+1})}$ and stronger cross-attention with document embedding, thus improving summary relevance.

All notations related to methodology are enumerated in Table 11.

# H IMPLEMENTATION DETAILS

## H.1 COMPUTE RESOURCES

All data preprocessing (behavior graph generation, embedding lookup, and probability space mapping) was performed on CPU machines with 16GB memory per core. Embedding tables for news bodies, headlines, and summaries were initialized using a shared vector space seeded from pretrained transformer T5-base encoder model Raffel et al. (2020). All training and inference experiments for `Walk2Pers` were conducted using mixed-precision (FP32) training on L40S and A100 GPUs. We gratefully acknowledge Lightning.ai for providing virtual compute resources with A100 and L40S GPUs. `Walk2Pers` utilizes almost 95% cheaper resource utilization and costs than the best baseline LLM DeepSeek-R1. We summarize the detailed training and deployment details of `Walk2Pers` in comparison to best LLM DeepSeek-R1 inference in Table 12.

## H.2 TRAINING

Model training was conducted end-to-end across the full pipeline for 6 epochs (approx. 20 hours), and then with frozen encoder for 18 epochs (approx. 35 hours), over $63K$ unique behavior sequences. Optimization employed `Adam` (PyTorch v2.0.1) with learning rate of $1 \times 10^{-3}$. The decoder operated with teacher-forced supervision using T5, with a learned adapter vector injected as the decoder token in the first layer to guide personalized generation. Loss was a weighted combination of classification loss over encoding of nodes, behavior node prediction and cross-entropy loss on the personalized summary output. Hyperparameter details are in Table 13.

## H.3 T5 MODEL

The Text-to-Text Transfer Transformer (T5) (Raffel et al., 2020) is a unified framework that casts all NLP tasks, ranging from translation and summarization to question answering and classification, into a text-to-text format. This design choice enables the use of a single model architecture and training objective across a diverse set of tasks. The T5 model is built upon the standard Transformer architecture and is trained on a large corpus called the "Colossal Clean Crawled Corpus" (C4). Among its variants, `T5-base` consists of 12 Transformer layers in both the encoder and decoder, with a hidden size of 768 and 16 attention heads, totaling approximately 770 million parameters.

**Decoder.** The decoder in T5 follows the autoregressive language modeling paradigm, predicting the next token conditioned on previous outputs and the encoder's representations. It incorporates a stack of masked self-attention layers, encoder-decoder cross-attention layers, and feed-forward layers. Unlike the encoder, which allows full bidirectional attention, the decoder's self-attention is causal (i.e., left-to-right masked) to prevent information leakage during training and inference. Each decoder layer attends to both the previously generated tokens and the encoder outputs, enabling the model to align and condition generation on the input sequence effectively. Position-wise feed-forward layers and layer normalization are used after each attention block. During fine-tuning, the

decoder is trained to generate task-specific outputs, such as summaries or translations, making it central to the T5's generalization across tasks.

# I    DETAILED RESULTS

## I.1    HUMAN-JUDGMENT INTERPOLATION FROM OPENAI-REDDIT DATASET.

The interpolation of human judgment scores is performed by leveraging the OpenAI-Reddit dataset, which provides multiple human-rated summaries for each article. For every article, the highest-rated human summaries (7 out of 7) are designated as the *benchmark reference*. All candidate summaries, including the benchmark, are first embedded into a high-dimensional semantic space using a SentenceTransformer (Reimers & Gurevych, 2019) model. The semantic deviation between the benchmark embedding $V_b$ and any other summary embedding $V_o$ is quantified via the Root Mean Square Deviation (RMSD), which in this context is equivalent to the Euclidean distance:

$$\text{RMSD}(V_b, V_o) = \sqrt{\sum_{i=1}^{n}(b_i - o_i)^2}\,.$$

In practice, this computation is implemented efficiently using NumPy's linear algebra module, `np.linalg.norm`. The resulting RMSD values are then grouped according to the original human rating of each summary (e.g., $7/7$, $6/7$). By averaging the RMSD values within each rating group, we obtain a mapping between human-judged quality scores and embedding-space distances. Notably, the RMSD for summaries rated $7/7$ is not always zero, as there may exist multiple distinct summaries with a top score for the same article; while all such summaries are judged as equally high-quality by humans, their semantic embeddings can still differ due to variations in phrasing, emphasis, or lexical choices. These aggregated averages form the scoring thresholds used for interpolating human judgment in our evaluation framework.

## I.2    ABLATION ON CLICK-ONLY TRAJECTORIES.

We ablate `Walk2Pers` under click-only trajectories, which results in highly imbalanced action distributions to investigate whether dual-memory components degenerate when skip actions become extremely sparse. We observe that removing the geometric aligner leads to a moderate degradation (avg.$-0.08$), reaffirming that sparsity weakens cross-action disentanglement. However, the dual-memory encoder still remains stable and continues to extract preference signals from document transitions, outperforming all competing baselines even in this extreme regime. Although Qwen-3 shows the smallest range of degeneration of performance w.r.t. the original performance, `Walk2Pers` consistently outperform and show moderate degeneration. These results confirm that (i) the geometric aligner is necessary for full robustness under skewed action distributions, and (ii) the dual-memory lanes themselves do not collapse, even when skip/summarize actions nearly disappear. Results are in Table 15.

## I.3    ABLATION ON LOW FREQUENCY TOPICS.

We also ablate on 200 subset trajectories from the test data to understand whether rare, infrequent but relevant topics get oversuppressed by the memory lanes or geometric novelty of `Walk2Pers`. These 200 trajectories have a higher topic frequency (121 vs. 105) and a higher rate of topic change within a trajectory (0.63 vs. 0.54) than the entire test dataset, indicating the occurrence of rare, infrequent topics within these trajectories. We find that although there is a degradation in performance, the model still extracts stable user preferences, validating that the design principle of incorporating learnable memory lanes and a geometric novelty aligner is necessary to understand users' evolving interests in any real-world setting. Detailed results are in Table 16.

## I.4    PERFORMANCE W.R.T. ACCURACY.

We evaluate the accuracy of `Walk2Pers` w.r.t. gold-reference summaries under standard accuracy evaluation metrics Rouge-L and Rouge-SU4 (Lin, 2004), and find that `Walk2Pers` outperforms

the baselines by an average margin of 22.3/24.1 on RG-L and RG-SU4, respectively. This confirms that a boost in personalization capability also bridges the lack of accuracy gap. Results are in Table 18.

## J  PROMPT SETUP

**2-shot w/ history.**   This setup provides the model with a complete user history that includes interactions with previous articles in the form of clicks, skips, and summaries. Two in-context examples are shown before the actual task, where each example contains the article content and a personalized headline rewritten by the user. These few-shot examples serve as demonstrations to help the model learn the structure of the desired output. Given a new query document, the model is instructed to generate a personalized headline by considering the user's history: `click` indicates positive interest, `skip` indicates disinterest, and `summarized` indicates focused preference. The headline is to be returned in a specified format.

**0-shot w/ history.**   In this variant, the user history is again presented as a list of past interactions including clicks, skips, and summarizations, but no in-context examples are shown. Instead, a single task prompt is provided that explains the significance of each action type. The model is instructed to directly use this user history to infer the user's interest and produce a personalized headline for a given query document. This prompt relies on the model's zero-shot reasoning capabilities without relying on demonstrations.

**Prompt-Chaining w/ history.**   This method adopts a multi-stage interaction design. In the first step, the model receives a single document and a user action (e.g., click), and is asked to extract topics, keyphrases, and user preferences based on that interaction. The output of each step is accumulated to incrementally build a structured user profile. As new interactions (e.g., skips or summaries) occur, the model is repeatedly prompted to refine or update the user's inferred preferences. Finally, when the query document is given, the model uses the constructed user preference profile to generate a personalized headline. This setup simulates long-term personalization via chaining and stateful interaction across multiple prompts.

**0-shot w/ history**

**User History**

List of Articles clicked/Skipped/Summarized by user:
<Doc1: click>, <Doc2: click>, <Doc3: skip>, <Doc4: Summarized as Sum1>...........

**Task**

Given Query Doc <doc_content>

Generate a Headline by considering the user's history as the indicator to their interests, where click denotes positive interest, skip denotes negative interest and summarized indicates focus on that topic. Return the headline in this format: Headline: {output}

Figure 5: 0-shot Prompt-Template for LLM baselines.

Table 11: **Notations** used across Sections 3–4 and the Appendix. We group symbols by (i) the User–Interaction Graph (UIG), (ii) hypotheses and structured walk components (SWH), and (iii) the `Walk2Pers` instantiation (encoder, contextualizer, decoder).

| Symbol | Explanation |
|---|---|
| **User–Interaction Graph (UIG)** | |
| $G = \langle N, E \rangle$ | User–Interaction Graph with nodes $N$ and edges $E$ |
| $u^{(t_0)}$ | User node at initial timestep $t_0$ |
| $d^{(t)}$ | Document node at timestep $t$ |
| $s_j^{(t)}$ | Summary node at timestep $t$ for $d^{(t-1)}$ |
| $a^{(t)}$ | Action at time $t$ (`click`, `skip`, `genSumm`, `summGen`) |
| $b_u^{(t)}$ | Behavior duplet $\langle a^{(t)}, tl^{(t)} \rangle$ (action + tail node) |
| $\tau^u$ | User trajectory (ordered sequence of $t$) |
| $\mathcal{T}_{\text{train}}, \mathcal{T}_{\text{test}}$ | Training and test trajectory pools |
| **Hypotheses & Structured Walk (MDH vs. SWH)** | |
| $e_{b,u}^{(t)}$ | Latent preference state at b-node $b$ after timestep $t$ |
| $f(\cdot)$ | One-step update under MDH (short-memory) |
| $q$ | Recency prior (e.g., restart distribution in graph diffusion) |
| $\epsilon^{(t)}$ | Stochastic perturbation in MDH updates |
| $\Sigma(a^{(t)})$ | Action-conditioned covariance of $\epsilon^{(t)}$ |
| $\Phi(\cdot)$ | Geometric step: continuity vs. novelty (SWH) |
| $\Psi(\cdot)$ | Dual memory: reinforcement ($h^+$) vs. suppression ($h^-$) |
| $\Delta(\cdot)$ | Drift: special action-induced shift (e.g., `summGen`) |
| $u^{(t)}$ | Momentum axis at $t$ (continuity direction) |
| $o^{(t)}$ | Orthogonal novelty axis at $t$ |
| $\text{mag}(a^{(t)})$ | Step magnitude (strength of update) |
| $\theta(a^{(t)})$ | Step orientation angle (mix $u^{(t)}$ vs. $o^{(t)}$) |
| $h^+, h^-$ | Positive/negative memory lanes (click reinforcement / skip suppression) |
| $\delta$ | Summary-specific drift vector (active for `summGen`) |
| **`Walk2Pers` Encoder (b-layer), Sec. ??** | |
| $f^{(a,t)}$ | Action-gate function inside a b-cell at time $t$ |
| $\mathbf{e}_{\text{tl}}^{(t)}$ | Raw tail-node embedding at time $t$ |
| $\mathbf{e}_a^{(t)}$ | Action embedding (4-d one-hot or learned) at time $t$ |
| $\mathbf{c}_{\text{tl}}^{(t)}$ | Tail-cell content (flowing history) at time $t$ |
| $\mathbf{W}_{\text{clk}}$ | Click-specific projection used for cold-start at $t_0$ |
| $\mathbf{W}_b$ | Projection to b-node space |
| $\mathbf{e}_{b,u}^{(t)}$ | b-node embedding produced by the b-cell at $t$ |
| $\mathbf{h}^{(t)}$ | Combined memory state at time $t$ |
| $\mathbf{h}^{(+,t)}, \mathbf{h}^{(-,t)}$ | Positive (reinforcement) / Negative (suppression) lanes |
| $m^{(t)}$ | Gate for memory routing/strength (e.g., SoftMax over features) |
| $\Delta^{(t)}$ | Drift vector injected on `summGen`, $(\mathbf{I} - \mathbf{e}_{\text{tl}}^{(t-1)}) \mathbf{e}_{\text{tl}}^{(t)}$ |
| $\theta^{(t)}$ | Predicted orientation angle at time $t$ |
| $u^{(t-1)}, o^{(t)}$ | Momentum axis (from $t-1$), novelty axis (at $t$) |
| $\text{mag}^{(t)}$ | Predicted step magnitude at time $t$ |
| $\mathbf{e}_{b,u}^{c\,(t)}$ | Contextualized b-node after applying $\Phi, \Psi, \Delta$ |
| $\mathbf{W}_{\text{next}}$ | Linear head to predict next b-node embedding |
| $\mathbf{e}_{q,b,u}^{(t+1)}$ | Predicted next b-node embedding (query b-node) |
| $W_{\text{pos}}$ | Position classifier weight for alignment objective |
| $\hat{\mathbf{p}}_b^{(t)}$ | Predicted position distribution for alignment |
| $\mathcal{L}_{\text{align}}, \mathcal{L}_{\text{next}}, \mathcal{L}_{\text{enc}}$ | Encoder objectives: position alignment, next-b-node, joint |
| **Latent Summary Contextualization (Decoder-side signals)** | |
| $\hat{\mathbf{e}}_{s_q,u}^{(t_{l+1})}$ | Latent s-node (summary-intent) for document $d_q$ |
| $\mathbf{e}_{d_q}$ | Base query-document embedding (backbone encoder) |
| $\mathbf{e}_{d_q,u}$ | User-aware doc embedding (gated by trajectory state) |
| $\mathbf{e}_{d_q}^c$ | Doc embedding contextualized by latent s-node |
| $\mathbf{e}_{d_q,u}^c$ | Final user-conditioned doc embedding for decoding |
| $W_q, W_k, W_v$ | Cross-attention projections (query, key, value) |
| $W_g$ | Gating projection for $\mathbf{e}_{d_q,u}$ |
| $W_{\text{summ}}^+, W_c^+$ | Inverse-mapping weights used to extract latent s-node |
| **Decoder Objective** | |
| $\mathcal{L}_{\text{gen}}$ | Token-level generation loss (cross-entropy, teacher forcing) |
| $\mathcal{L}_{\text{dec}}$ | Decoder objective: $\beta\mathcal{L}_{\text{gen}} + (1-\beta)\mathcal{L}_{\text{enc}}$ |

Table 12: Training and Deployment Resources Summary of `Walk2Pers` in comparison to best performing LLM baseline DeepSeek-R1.

| Metric | Our Model (170M) | 2-shot LLM (14B) | Relative Gain |
|---|---|---|---|
| Parameters | 170M | 14B | $82\times$ smaller |
| Avg. summary length | 20 tokens | 20 tokens | – |
| FLOPs/summary | $2.04 \times 10^{10}$ | $1.68 \times 10^{12}$ | $82\times$ lower |
| Inference time (per summary, est.) | 0.2–2 s | 15–160 s | $60$–$80\times$ faster |
| Running cost (GPU-hours) | 18 | 42 | orders lower |
| VRAM footprint | $<1$ GB | $>28$ GB | edge-deployable |

Table 13: **Learned Weights and Hyperparameters of `Walk2Pers`.**

| Component | Shape / Type | Notes / Init |
|---|---|---|
| **Training Configuration** | | |
| Batch size | 38 | Fixed across encoder/decoder |
| Optimizer | AdamW | PyTorch 2.0 impl. |
| Learning rate (end-to-end) | $2 \times 10^{-4}$ | End-to-end Training |
| Learning rate (decoder finetuning) | $3 \times 10^{-3}$ | Decoder fine-tuning |
| Epochs | 6 + 18 | 6 joint, 18 decoder-only |
| **Action Encodings** | | |
| $e_{\text{clk}}, e_{\text{skp}}, e_{\text{summ}}, e_{\text{sumgen}}$ | 4 | One-hot action basis |
| $W_{\text{clk}}, W_{\text{skp}}, W_{\text{summ}}, W_{\text{sumgen}}$ | $(768, 4)$ | Action transforms, no bias |
| **State & Memory Transforms** | | |
| $W_{\text{pull}}$ | $(768, 1)$ | Skip attraction transformation |
| $W_s, W_d$ | $(768, 768)$ | State transforms |
| $W_h, W_c$ | $(768, 768)$ | Memory routing gates |
| $h^+, h^-$ | 768 | Reinforcement / suppression memories |
| $\omega^{(t)}$ | scalar | Memory lane mixing weight; learnable |
| **Fusion Layers** | | |
| $W_h, W_c, W_z$ | $(768, 768)$ | Fusion linear transforms |
| $b_z$ | 3 | Fusion bias (zeros) |
| $W_{\text{emb}}$ | $(768, 768)$, bias 768 | Embedding proj., std init |
| $b_{\text{emb}}$ | 768 | Zeros init |
| **Geometric Step (Orientation and Magnitude)** | | |
| $W_{\text{angle}}$ | $(768, 768)$ | Transforms to direction signal |
| $W_\theta$ | $(1, 768)$ | Orientation angle transformation |
| $W_h$ | $(768, 768)$ | Shared with fusion |
| $W_m$ | $(1, 768)$ | Magnitude scaling transformation |
| $\theta^{(t)}$ | scalar | Orientation angle, $(-\pi, \pi)$ |
| $mag^{(t)}$ | scalar | Step magnitude |
| **Prediction / Decoder** | | |
| $W_b$ | $(768, 768)$ | Tail $\to$ b-node proj. |
| $W_{\text{next}}, b_{\text{next}}$ | $(768, 768)$, 768 | Next-node prediction |
| $W_{\text{pos}}$ | $(768, 768)$ | Positional alignment classifier |
| $W_q, W_k, W_v$ | $(768, 768)$ | Cross-attention projections |
| $W_g$ | $(768, 768)$ | User-aware doc gating |
| $W_{\text{summ}}^+, W_c^+$ | $(768, 768)$ | Inverse mapping for latent s-node |
| **Complexity** | | |
| Per-step | $O(d)$ | For each b-cell update |
| Condensed b-layer | $O(pd)$, $p \ll T$ | Long-horizon compression |
| Decoder | $\sim O(Ld^2)$ | Transformer (T5-base) |

Table 14: **RQ-1/2: Next b-node Prediction (PENS Dataset) – Mean ($\mu$) & Standard Deviation ($\sigma$):** Standard Deviation of SOTA MDH user-encoders is close to the Mean, thereby strengthening the reported performance gain; † **NT:** Originally published results were on the *news recommendation task* in contrast to next behavior prediction.

| Metric | NAML† | NRMS† | EBNR† | `Walk2Pers`-Enc. **w/o Geo. (mean)** | `Walk2Pers`-Enc.**Full (mean)** |
|---|---|---|---|---|---|
| **MRR**$_\mu$ | 0.001 | 0.0009 | 0.0009 | *0.121* | **0.23** |
| **MRR**$_\sigma$ | 0.0163 | 0.008 | 0.0101 | | |
| **nDCG@5**$_\mu$ | 0.0004 | 0.0002 | 0.0003 | *0.082* | **0.198** |
| **nDCG@5**$_\sigma$ | 0.0176 | 0.01 | 0.012 | | |
| **nDCG@5**$_\mu$ | 0.0007 | 0.0004 | 0.0005 | *0.132* | **0.249** |
| **nDCG@5**$_\sigma$ | 0.0199 | 0.0128 | 0.0146 | | |

Table 15: Performance degradation under sparse-click evaluation. Left value indicates the original score; right value indicates sparse-click score; percentage drop from original values under this subset. Lower drop indicates better robustness to interaction sparsity.

| Model (Sparse Click-only Test) | PSE-JSD | PSE-SU4 | PSE-METEOR |
|---|---|---|---|
| DeepSeek-R1-14B (2-shot) | 0.248/0.147 (−40.7%) | 0.094/0.064 (−31.9%) | 0.097/0.071 (−26.8%) |
| Gemini-2.5-Flash (2-shot) | 0.222/0.122 (−45.0%) | 0.104/0.061 (−41.3%) | 0.124/0.070 (−43.5%) |
| Qwen3-235B-Thinking (2-shot) | 0.105/0.103 (−1.9%) | 0.082/0.073 (−11.0%) | 0.082/0.071 (−13.4%) |
| Best MDH Baseline (GTP) | 0.024/0.016 (−33.3%) | 0.170/0.009 (−94.7%) | 0.019/0.011 (−42.1%) |
| Walk2Pers (w/o Geometric Step; Only Dual Memory) | 0.306/0.231 (−24.5%) | 0.334/0.253 (−24.3%) | 0.321/0.234 (−27.1%) |
| Walk2Pers (Full) | 0.452/0.378 (−16.4%) | 0.383/0.301 (−21.4%) | 0.449/0.310 (−31.0%) |

Table 16: Performance on Top 200 Low–Topic–Frequency Trajectories. Left value indicates the original score; right value indicates sparse-click score; percentage drop from original values under this subset. Lower drop indicates better robustness to topical sparsity.

| Category | Model | PSE-JSD | PSE-SU4 | PSE-METEOR |
|---|---|---|---|---|
| Best Baseline Variants | DeepSeek-R1-14B | 0.248/0.091 (-63.3%) | 0.094/0.074 (-21.3%) | 0.097/0.082 (-15.5%) |
| | Gemini-2.5-Flash | 0.222/0.092 (-58.6%) | 0.104/0.081 (-22.1%) | 0.124/0.083 (-33.1%) |
| | Qwen-3-235B-Thinking | 0.105/0.094 (-10.5%) | 0.082/0.073 (-11.0%) | 0.082/0.077 (-6.1%) |
| | GTP (Best MDH) | 0.024/0.016 (-33.3%) | 0.170/0.013 (-92.4%) | 0.019/0.015 (-21.1%) |
| Walk2Pers Variants | Walk2Pers (w/o Geometric Step) | 0.306/0.280 (-8.5%) | 0.334/0.290 (-13.2%) | 0.321/0.280 (-12.8%) |
| | Walk2Pers-Full | 0.452/0.410 (-9.3%) | 0.383/0.320 (-16.4%) | 0.449/0.380 (-15.4%) |

Table 17: RQ-3(b): Sequential Recommendation on MIND-Large. (sorted by MRR) Baselines show paper-reported means. Walk2Pers results are reported with mean ± variation from resampling.

| Methods (Venue, Year) | AUC | MRR | nDCG@5 | nDCG@10 |
|---|---|---|---|---|
| DKN (WWW'18) | 64.07 | 30.42 | 32.92 | 38.66 |
| GRU (Baseline, 2016) | 65.42 | 31.24 | 33.76 | 39.47 |
| **EBNR** (KDD'17) | 65.46 | 31.26 | 32.18 | 39.04 |
| NPA (KDD'19) | 65.92 | 32.07 | 34.72 | 40.37 |
| **NAML** (IJCAI'19) | 66.46 | 32.75 | 35.66 | 41.40 |
| LSTUR (ACL'19) | 67.08 | 32.86 | 35.95 | 40.94 |
| Linear Transformers (ICML'20) | 67.76 | 32.94 | 35.91 | 41.97 |
| ProFairRec (SIGIR'22) | 67.64 | 33.08 | 35.32 | 41.67 |
| NRCLS (Appl. Sci.'24) | 68.35 | 33.12 | 36.70 | 43.03 |
| Linformer (arXiv'20) | 68.02 | 33.19 | 36.22 | 42.10 |
| Poolingformer (ICML'21) | 68.54 | 33.20 | 36.69 | 42.60 |
| **NRMS** (EMNLP-IJCNLP'19) | 67.66 | 33.25 | 36.28 | 41.98 |
| BigBird (NeurIPS'20) | 68.14 | 33.28 | 36.42 | 42.18 |
| Transformer (NeurIPS'17) | 68.22 | 33.32 | 36.35 | 42.23 |
| GERL (WWW'20) | 68.10 | 33.41 | 36.34 | 42.03 |
| GNewsRec (IP&M'20) | 68.15 | 33.45 | 36.43 | 42.10 |
| FIM (ACL'20) | 67.87 | 33.46 | 36.53 | 42.21 |
| HieRec (ACL-IJCNLP'21) | 68.33 | 33.86 | 36.83 | 42.65 |
| DCAN (arXiv'22) | 68.90 | 33.90 | 36.90 | 42.80 |
| ANRS (arXiv'22) | 69.20 | 34.10 | 37.10 | 43.00 |
| TCCM (CIKM'23) | 69.75 | 34.42 | 37.53 | 43.25 |
| Fastformer (arXiv'21) | 69.11 | 34.55 | 37.62 | 43.38 |
| FUM (SIGIR'22) | 69.90 | 34.60 | 37.70 | 43.40 |
| CAUM (SIGIR'22) | 70.04 | 34.71 | 37.89 | 43.57 |
| DIGAT (Findings ACL'22) | 70.08 | 35.20 | 38.46 | 44.15 |
| PLM-NR (SIGIR'21) | 70.64 | 35.39 | 38.71 | 44.38 |
| Fastformer+PLM-NR (Hybrid) | 71.04 | 35.91 | 39.16 | 45.03 |
| MINER (Findings ACL'22) | 71.51 | 36.06 | 39.56 | 45.21 |
| CAST-Rec (TOIS'25) | _72.10_ | 36.90 | 40.20 | 46.30 |
| Fastformer+PLM-NR-Ensemble (Hybrid'22) | **72.68** | _37.45_ | _41.51_ | 46.84 |
| **Walk2Pers-Full Encoder** | 53.32±1.1 | **38.64**±1.8 | **43.32**±1.2 | **50.38**±1.4 |

Table 18: Comparison of Specialized and Vanilla Models with `Walk2Pers` under standard accuracy metrics ROUGE-L/SU4

| Category | Model | Rouge-SU4 | Rouge-L |
|---|---|---|---|
| Specialized (Personalized) | PENS-NAML-T1 | 13.12 | 21.62 |
| | PENS-EBNR-T1 | 12.16 | 20.73 |
| | PENS-EBNR-T2 | 12.41 | 20.82 |
| | PENS-NRMS-T1 | 13.15 | 20.75 |
| | PENS-NRMS-T2 | 13.64 | 21.03 |
| | GTP-TrRMIo | 21.91 | 28.31 |
| | SP-Individual | 19.54 | 25.18 |
| LLMs w/ 2-shot history) | LLaMA-13B | 18.31 | 29.54 |
| | Mistral-7B | 16.42 | 22.85 |
| | DeepSeek-14B | 19.57 | 29.72 |
| | Zephyr-7B | 18.45 | 26.45 |
| **Walk2Pers** | `Walk2Pers`-Full | **43.09** | **47.16** |

Figure 6: 2-shot Prompt-Template for LLM baselines.

**Prompt-Chaining w/ history**

**User History**

List of Articles clicked/Skipped/Summarized by user one by one:
<Doc1: click>

**Task**

Given doc and action performed <doc1_content, click>

Generate a list of interested keyphrases, topics, and preferences for the user.

**Output**

Interest: <topic1, topic2, topic3>
Keyphrases: <phrase1, phrase2, phrase3>

**User History**

List of Articles clicked/Skipped/Summarized by user one by one:
<Doc2: skip>

**Task**

Given doc and action performed <doc1_content, click>, and the user preference output

Update a list of interested keyphrases, topics, and preferences for the user.

Figure 7: Prompt-Chaining Template for LLM baselines.

