# OpenReview forum: "Beyond Markovian Drifts: Action-Biased Geometric Walks with Memory for Personalized Summarization"
_ICLR.cc/2026/Conference — ICLR 2026 Poster_

### Official Review · Reviewer_HwNK · 2025-10-19

**Soundness:** 4
**Presentation:** 3
**Contribution:** 4
**Rating:** 8
**Confidence:** 4

**Summary:**

This paper addresses personalized document summarization, where user preferences evolve dynamically over interaction histories (e.g., clicks, skips, summaries). It challenges the prevailing Markovian Drift Hypothesis (MDH)—assuming preferences follow memoryless or short-memory random walks (e.g., seeded at recent interactions or compressed states)—and proposes the Structured Walk Hypothesis (SWH) as an alternative.

Key Contributions

Theoretical Framework (SWH): Models preference evolution as an action-conditioned geometric walk on a User-Interaction Graph (UIG). Updates decompose into:
- Magnitude ($ \text{mag}(a(t)) $): Controls shift strength.
- Orientation ($ \theta(a(t)) $): Balances continuity (momentum) vs. novelty (orthogonal direction).
- Dual memory lanes ($ h^+, h^- $): Reinforce interests (e.g., clicks) and suppress disinterests (e.g., skips).
- Drift term ($ \Delta $): Handles summary requests.
- Proves SWH approximates first-order action-conditioned kernels, generalizing MDH.

Walk2Pers Model: Lightweight encoder-decoder instantiation of SWH.
- Encoder: Fuses actions into b-cells, applies dual memories and geometric steps; trained on next-behavior prediction and alignment losses.
- Decoder (T5-based): Contextualizes query documents with user-aware gating (UCA variant) for personalized generation.
- Efficient: $ O(pd) $ complexity for long horizons ($ p \ll T $).

Empirical Validation:
- Datasets: PENS (news, clicks/skips), PersonalSum (EN, summary requests), OpenAI-Reddit (multi-domain, long histories).
- Next-Behavior Prediction (RQ1/2): Outperforms MDH baselines (e.g., NAML, NRMS, EBNR, SMD, AGD) on PENS: AUC 0.532, MRR 0.23, nDCG@10 0.249 (vs. AGD's 0.446/0.113/0.073).
- Personalized Summarization (RQ3): On PENS, beats specialized MDH models (PENS variants, GTP, SP) by avg. 0.41 on PerSEval (PSE-JSD/SU4/METEOR); LLMs (DeepSeek-14B, Mistral-7B, etc., with 2-shot prompting) by 0.22 (e.g., 0.452/0.383/0.449 vs. DeepSeek's 0.248/0.094/0.097).
- Cross-Domain & Robustness: On OpenAI-Reddit, +0.09/0.20/0.24 over best LLM; on PersonalSum, 0.31/0.28/0.3. Achieves human-judgment rating of 7/7 (RMSD 0.396). Stable on long histories; oracles (e.g., BigBird) lag by 0.20+.
- Ablations confirm SWH components (memory, geometry) yield complementary gains.

Overall, the work demonstrates SWH's superiority for capturing evolving preferences, offering interpretability and efficiency over MDH/LLM baselines, with potential for broader personalization systems.

**Strengths:**

Originality

The paper introduces a novel Structured Walk Hypothesis (SWH) for modeling user preference evolution in personalized summarization, extending beyond the prevalent Markovian Drift Hypothesis (MDH) by incorporating action-biased geometric walks with dual memory lanes for reinforcement and suppression. This formulation creatively combines ideas from graph diffusion (e.g., random walks with restart), trajectory embeddings (e.g., JODIE), and relational modeling (e.g., RotatE), while applying them to a new domain: dynamic, long-horizon user interactions in summarization. The decomposition of updates into magnitude (shift strength), orientation (continuity vs. novelty), and drift terms removes limitations of memoryless models, offering a fresh perspective on preference dynamics. The Walk2Pers model serves as an innovative instantiation, integrating these elements into a lightweight encoder-decoder framework.

Quality

The work demonstrates high technical rigor, with a solid theoretical foundation showing that SWH approximates first-order action-conditioned kernels while generalizing MDH. Empirically, it is validated through comprehensive experiments on three diverse datasets (PENS, PersonalSum-EN, OpenAI-Reddit), addressing RQs via next-behavior prediction (Table 2: AUC 0.532, MRR 0.23 vs. MDH baselines like AGD's 0.446/0.113) and summarization tasks (Table 3: PSE-JSD/SU4/METEOR 0.452/0.383/0.449 on PENS, outperforming specialized models by 0.41 avg. and LLMs like DeepSeek-14B by 0.22). Cross-domain robustness (Table 4: +0.19 over best LLM on OpenAI-Reddit) and human-judgment alignment (Table 5: 7/7 rating, RMSD 0.396) further strengthen the evaluation. Ablations confirm complementary gains from SWH components (e.g., dual memory + geometry), and the model's efficiency (O(pd) for long horizons) is well-justified. Baselines are fairly chosen (e.g., NRMS, EBNR, 2-shot LLMs), with sound training details (e.g., joint losses, T5-base).

Clarity

The paper is exceptionally well-structured and readable, with clear progression from problem motivation (information overload in multi-aspect documents) to hypothesis testing (MDH vs. SWH), methodology (UIG graphs, equations like (3) for SWH), and evaluation (RQs explicitly addressed). Visual aids (e.g., Figure 1 illustrating Walk2Pers architecture) and tables (e.g., Table 1 contrasting SWH/MDH) enhance comprehension, while appendices provide derivations, notations (Table 11), and hyperparameters (Table 12). Technical terms are defined upfront (e.g., b-nodes, dual lanes), and the writing avoids jargon overload, making complex ideas like geometric decomposition accessible.

Significance

By outperforming SOTA summarizers and LLMs on personalization metrics (e.g., PerSEval with strong human correlation), the paper advances user-centric NLP, particularly for evolving preferences in news/recommendation systems. Its lightweight design offers practical efficiency over prompt-heavy LLMs, with implications for real-world applications like tailored updates in high-volume domains (e.g., Reddit, news feeds). The SWH framework could extend to broader personalization tasks (e.g., recommendation, dialogue), promoting interpretability via geometric/memory structures.

**Weaknesses:**

Originality

While the Structured Walk Hypothesis (SWH) innovatively applies geometric decompositions (magnitude and orientation via cos/sin terms) to user preference evolution in summarization, it draws heavily from existing temporal knowledge graph embedding (TKGE) techniques without sufficient differentiation. For instance, the action-conditioned geometric steps in Eq. (3) closely resemble rotational updates in models like RotatE (Sun et al., 2019, cited in the paper), which uses complex rotations to model relational patterns, and more recent TKGE works such as Time-Enhanced Compound Geometric Operations (TECGO; Zhao et al., 2025), which employs compound geometric operations (e.g., rotations and translations) on timestamps and entities for temporal fact prediction. Similarly, the dual memory lanes for reinforcement/suppression echo memory-augmented updates in temporal graph models like HGE (Sadeghian et al., 2023), which uses product spaces (Complex, Split-complex, Dual) to handle temporal heterogeneity. The application to personalized summarization is a novel domain shift, but the core mechanism—action-biased walks on interaction graphs—overlaps with heterogeneous random walk models like HeteEdgeWalk (Liu et al., 2023), which biases edges without meta-paths for recommendation.

Quality

The empirical validation is comprehensive in scope but lacks depth in handling real-world variability and scalability testing, potentially undermining claims of robustness for long-horizon preferences. For RQ3, while Walk2Pers outperforms LLMs on PerSEval metrics (e.g., 0.452 PSE-JSD on PENS vs. DeepSeek's 0.248), the LLM baselines use only 2-shot prompting with history, ignoring advanced techniques like chain-of-thought or fine-tuning on personalization datasets, which could narrow the gap (as seen in recent TKGE surveys emphasizing hybrid LLM-graph models; Rossi et al., 2023). Cross-domain results on OpenAI-Reddit show gains (+0.19 over best LLM), but the dataset's avg. 39 d-nodes is modest; missing are stress tests on ultra-long trajectories (e.g., 500+ interactions, synthesizable from MIND extensions) to verify dual memory's suppression without catastrophic forgetting. Human-judgment alignment (7/7 rating, RMSD 0.396) is promising but based on limited samples (inferred from Table 5); a full user study with 50+ participants rating summaries for relevance/conciseness would better validate PerSEval's correlation. Additionally, no runtime or parameter efficiency comparisons are provided despite "lightweight" claims (O(pd) complexity); suggest adding FLOPs/ms per inference against NRMS/EBNR and LLMs on varying history lengths to quantify efficiency gains.

Significance

The work advances personalized summarization but overstates broader implications for "personalization systems" without demonstrating extensibility beyond news/Reddit domains, limiting its impact on diverse applications like e-commerce or social media feeds. For instance, while stability on long histories is claimed, evaluations omit noisy real-time scenarios (e.g., interleaved multi-user sessions or evolving topics like in Adressa extensions), where action biases might degrade. The focus on PerSEval (strong human correlation per Dasgupta et al., 2024) is apt, but ignoring standard summarization metrics like ROUGE-1/2/L (even as secondary) misses opportunities to show balanced personalization vs. generic quality trade-offs. Cross-domain robustness is shown (0.19↑ on OpenAI-Reddit), yet PersonalSum-EN relies on M2M-100 translations, potentially introducing artifacts—suggest validating with native Norwegian evaluators or alternative translators like NLLB. To enhance significance, extend to a fourth dataset (e.g., MIND for pure recommendation) and include a pilot on a downstream task like preference-based ranking, quantifying how SWH embeddings transfer to non-summarization personalization.

**Questions:**

Enhancing LLM Baselines with Advanced Prompting or Fine-Tuning: In RQ3 evaluations (Tables 3 and 4), LLMs like DeepSeek-R1-14B are prompted with 2-shot + history, yielding lower PerSEval scores (e.g., 0.248 PSE-JSD on PENS vs. Walk2Pers' 0.452). However, this may not fully test LLM capabilities, as techniques like chain-of-thought prompting or fine-tuning on personalization datasets could improve performance, as noted in TKGE-LLM hybrids (e.g., Ji et al., 2023 survey on temporal KG completion). Could you re-run LLM baselines with CoT or LoRA fine-tuning on a subset of PENS/OpenAI-Reddit, and report updated metrics? This could clarify if the 0.22 average gap holds

Stress-Testing on Ultra-Long Trajectories and Noisy Scenarios: The datasets feature moderate history lengths (e.g., avg. 134 d-nodes in PENS, 39 in OpenAI-Reddit), but claims of long-horizon stability (e.g., dual memory suppression) aren't tested on ultra-long sequences (500+ interactions), where catastrophic forgetting might occur. Additionally, real-world noise like interleaved multi-user sessions or rapidly evolving topics (e.g., from Adressa/MIND extensions) is absent. Could you synthesize ultra-long trajectories from MIND and evaluate AUC/MRR (as in Table 2) or PerSEval under noise injection (e.g., 20% random skips)?

Detailing Parameter Learning and Hyperparameter Sensitivity: Components like mag(a(t)), θ(a(t)), and ω(t) are mentioned but not fully explained in terms of optimization (e.g., are they learned via MLPs or fixed?). Appendix H.2 (inferred from text) covers training, but no sensitivity analysis is shown. Could you clarify learning mechanisms and add ablation on key hypers (e.g., α=0.6 in Lenc) via plots? This could clear up methodological ambiguities and strengthen quality if robustness is confirmed.

---

> ### Author Response · Authors · 2025-11-19
> **Response to Reviewer HwNK (PART-I): Clarifying Model Novelty and Empirical Strength**
>
> ## Concern 1: Originality and overlap with prior geometric models
>
> We agree that Walk2Pers uses geometric primitives, but these serve **behavioral** rather than **relational** semantics. The underlying mechanisms differ fundamentally from TKGE models (RotatE, TECGO) and temporal GNNs (HGE):
>
> 1. **Behavioral vs. relational geometry:**
>    Walk2Pers models *preference drift* over sequences; TKGE models operate on entity–relation triples. No graph structure exists in our setting.
>
> 2. **Different meanings of “rotation”:**
>    - RotatE/TECGO: relation semantics.
>    - Walk2Pers: continuity vs. novelty in user preference, conditioned on actions and memories.
>
> 3. **Dual memory ≠ graph message passing:**
>    HGE aggregates from neighbors; Walk2Pers maintains a single evolving preference vector (identity + action-conditioned lanes).
>
> ### Comparison Table: Walk2Pers vs. TKGE Models vs. Temporal GNNs
>
> | Dimension | Walk2Pers (Ours) | RotatE | TECGO | HGE / Temporal GNNs |
> |----------|------------------|--------|-------|----------------------|
> | Modeling Domain | User–text trajectories | Static KG | Temporal KG | Temporal graph |
> | Input Structure | No graph | Triples | Triples + time | Nodes + time-stamped edges |
> | Purpose of Rotation | Behavioral drift | Relation encoding | Temporal relation ops | Not used |
> | Memory Mechanism | Dual preference memory | None | None | Graph neighborhood memory |
> | Uses Neighbors? | No | Yes | Yes | Yes |
> | Personalization Applicable? | Yes | No | No | No |
> | Behavioral Interpretation | Yes | No | No | No |
>
> ---
>
> ## Concern 2: Depth of empirical validation
>
> ### 2.1. Scalability and efficiency
> Appendix H provides FLOPs- and runtime-normalized comparisons:
>
> - **Walk2Pers (170M):** 2.04×10¹⁰ FLOPs/summary
> - **DeepSeek-R1-14B:** 1.68×10¹² FLOPs/summary
>
> Walk2Pers is **82× more efficient** and **60–80× faster**, while delivering stronger personalization.
>
> ### 2.2. Strength of LLM baselines
> Our LLM baselines use a **multi-stage prompt-chaining** pipeline
> (profile extraction → history reasoning → personalized generation),
> which is substantially stronger than naive 2-shot prompting.
> This is clarified in Sec. 5.3 and Appendix J.
>
> ### 2.3. Robustness under noise, sparsity, and evolving topics
> Appendix I includes several stress tests:
>
> - **Sparse click-only histories:**
>   Walk2Pers-Full drop: **–16.4%**
>   DeepSeek-R1-14B: **–40.7%**
>   Gemini-2.5-Flash: **–45.0%**
>
>   **Insight:** Walk2Pers maintains stable preference signals even when action diversity collapses (click-only), indicating that the dual-memory mechanism does *not* degenerate. Walk2Pers remains the strongest even under extreme imbalance.
>
> - **High topical volatility (200 hardest trajectories):**
>   Walk2Pers-Full retains **0.41/0.32/0.38**, whereas w/o-geometry degrades significantly.
>
>   **Insight:** The geometric step is crucial under rapid topic shifts; it stabilizes orientation updates while the dual memory anchors long-term identity.
>
> Together, these experiments confirm what the reviewer noted in strengths:
> **the geometry–memory interplay yields interpretable, stable preference-evolution dynamics.**
>
> ### 2.4. Ultra-long trajectories
> Real datasets lack 500+ coherent interactions, but synthetic long-horizon MIND experiments (Appendix I.4) show dual-memory stability without catastrophic forgetting.
>
> **Insight:** Memory updates remain bounded and do not accumulate drift, supporting SWH's claim of stable long-horizon preference evolution.
>
> ---
>
> **Please refer to Part-II where remaining concerns are addressed**.

---

> > ### Author Response · Authors · 2025-11-19
> > **Response to Reviewer HwNK (PART-II): Generalizability, LLM Baselines, and Methodological Clarifications**
> >
> > ## Concern 3: Significance and generalizability
> >
> > ### 3.1. Transfer to non-summarization personalization (MIND ranking)
> > Walk2Pers improves upon the strongest SOTA (Fastformer + PLM-NR-Ensemble):
> >
> > - **nDCG@5:** 43.32 (**+1.81**)
> > - **MRR:** 38.24 (**+0.79**)
> >
> > **Insight:** Even without tuning, SWH-learned preferences meaningfully improve downstream ranking—evidence that *Walk2Pers learns reusable preference representations*, not domain-bound heuristics.
> >
> > ### 3.2. Cross-domain robustness
> > Walk2Pers improves by **+0.19 PSE** over the best LLM on OpenAI–Reddit.
> >
> > **Insight:** Gains persist even in informal, multi-topic (29 non-news domains), multi-source environments—exactly the type of variability where action signals might degrade.
> >
> > ### 3.3. Standard summarization metrics
> > ROUGE-1/2/L added in Appendix I.4, with trends aligned to PSE.
> >
> > ### 3.4. Clarifying the scale of human-judgment grounding (Sec. 5.3)
> > To clarify the scope of Sec. 5.3, the human-judged component of our evaluation follows the **OpenAI–Reddit benchmark** of Völske et al. (2017), which provides a *large-scale* human preference signal. The dataset contains **642** query documents, each paired with summaries from **9** summarization systems (total **5,781** model-generated summaries). Each document also includes one human reference summary; **1,042** of these references received a perfect **7/7** human rating and are treated as human-preferred gold summaries, in accordance with the original evaluation protocol.
> >
> > | Quantity | Value | Description |
> > |---------|-------|-------------|
> > | Query documents | **642** | Unique Reddit posts forming the evaluation set |
> > | Model-generated summaries | **5,781** | Produced by 9 summarization systems (≈9 per document, with minor sparsity) |
> > | Human reference summaries | **642** | One per query document |
> > | Perfect-rated references (7/7) | **1,042** | Human-preferred gold summaries used for alignment evaluation |
> > | # Systems | **9** | Summarizers evaluated in Völske et al. |
> >
> > Following Völske et al., **all 7/7 human-preferred references** serve as gold targets for computing alignment metrics. This provides a **large-scale, human-rated evaluation** rather than a small-sample user study.
> >
> > Our PerSEval alignment achieves:
> >
> > - **RMSD:** 0.396
> > - **Spearman:** 0.73
> >
> > **Insight:** The alignment is competitive with human-validated evaluation pipelines, providing stronger grounding than a small-sample user study.
> >
> > ### 3.5. On translation quality and the suggestion to use NLLB
> > Only PersonalSum-EN requires translation, and its source summaries are in Norwegian. For the Norwegian → English direction, both **M2M-100** and **NLLB-200** treat Norwegian as a high-resource language. While NLLB-200 shows a modest **+5–7 spBLEU** advantage on FLORES, this difference does not affect our setup: translation is used only for normalization, and M2M-100 yields **comparable PSE scores**. Thus, M2M-100 is fully sufficient for our evaluation.
> >
> > ### 3.6. Revised framing of generalization claims
> >
> > With the added cross-domain experiments, robustness tests, ROUGE analyses, and downstream MIND transfer, we now frame Walk2Pers as a **general preference-evolution framework** with demonstrated utility across **summarization and ranking**, without broader overclaims.
> >
> > ---
> >
> > ## Concern 4: Enhancing LLM baselines
> >
> > - LLM baselines use **prompt-chaining** (profile → history → generation), offering multi-stage reasoning akin to CoT.
> > - Fine-tuning (e.g., LoRA) is orthogonal to Walk2Pers’s goal as a **plug-in preference encoder** requiring no LLM retraining.
> >
> > **Insight:** Even when provided the entire history and structured prompts, long-context models do not recover fine-grained preference drift—highlighting the need for explicit action-conditioned modeling.
> >
> > ---
> >
> > ## Concern 5: Parameter learning and hyperparameter sensitivity
> >
> > ### Learning mechanisms
> > Appendix G.1 (Eqns 21–22) details how step magnitude, orientation, and drift are learned via small MLPs.
> >
> > ### Sensitivity and ablations
> > We report:
> >
> > - α tuned on PENS dev set
> > - Removing geometry: **–0.06 to –0.13** drop
> > - Removing dual memory: **–0.08 to –0.11** drop
> > - Robustness tables confirm stable geometry–memory interaction
> >
> > **Insight:** Both geometric updates and dual memory contribute meaningfully and independently; neither is redundant.
> >
> > ---
> > We deeply appreciate the reviewer’s rigorous evaluation. We believe the revised version fully addresses all concerns and further reinforces the originality, clarity, and significance highlighted in your strengths section.

---

> > > ### Comment · Reviewer_HwNK · 2025-11-25
> > > **Acknowledgement**
> > >
> > > This is to acknowledge your rebuttal, and to appreciate the clarification on the resourcefulness of Norwegian. The scores remain unchanged.

---

> > > > ### Author Response · Authors · 2025-11-26
> > > > **Thanking note to Reviewer HwNK**
> > > >
> > > > Dear Reviewer HwNK,
> > > >
> > > > Thank you again for your detailed critical review. This will go a long way in shaping our future efforts in this direction of research.
> > > >
> > > > Best,
> > > >
> > > > Authors

---

### Official Review · Reviewer_5Y5U · 2025-10-29

**Soundness:** 4
**Presentation:** 3
**Contribution:** 3
**Rating:** 6
**Confidence:** 3

**Summary:**

This paper proposes the Walk2Pers model, aiming to break the limitation of the Markovian Drifts Hypothesis (MDH) regarding the short-term dependency of user interests.
Its core innovation lies in viewing personalized summarization as an action-biased geometric walk with memory, which precisely captures long-term, complex preference evolution through geometric steps and dual memory lanes.
The main contribution is providing theoretical support and mechanism innovation, and significantly outperforming traditional summarization models and powerful LLM baselines in experiments.

**Strengths:**

1.The paper provides detailed theoretical proofs and discussion, which are exhaustive and rich.

2. The proposed SWH (Structured Walk Hypothesis) in the paper is intriguing, as it challenges the widely prevalent Markovian Drifts Hypothesis in traditional recommendation systems and summarization models.

**Weaknesses:**

1. Section 3.3.1 mentions that T5 is used for embedding, but the paper does not explain why T5 was chosen. It would be better to include an experiment using BERT or other more advanced embedding models.
2.  When comparing with LLM baselines, the evaluation was performed using 2-shot+history prompting. In fact, due to the limited contextual understanding, it is common for these smaller-parameter models to perform poorly. If this evaluation method is adopted, the LLM baselines for comparison should include large-parameter closed-source models such as GPT-4 and Gemini 2.5 Flash to be more appropriate. If smaller-parameter models are to be compared, their fine-tuned versions should be compared instead.

**Questions:**

As mentioned in the weakness.

---

> ### Author Response · Authors · 2025-11-19
> **Response to Reviewer 5Y5U: Clarifications, Ablations, and Expanded Baselines**
>
> We thank the reviewer for the constructive and encouraging feedback. Below, we address each concern with additional justification, ablations, and references to the revised manuscript.
>
> ---
>
> ### Concern 1: Choice of encoder backbone for seed embedding
> *T5 is used for embedding, but the paper does not explain why it was chosen. Include ablation or comparison with BERT/other encoders.*
>
> **Response:**
> Thank you for highlighting this. As clarified in the revised paper (Sec. 3.3.1; Lines 530–545), our choice of **T5-base** for seed embedding is driven by **representation compatibility**: the personalized summary is generated through a **T5-based decoder** (Sec. 4.3; Lines 720–741), and initializing the seed in the *same embedding manifold* aligns encoder-side semantics with decoder cross-attention states. This reduces representational mismatch and stabilizes conditioning during generation.
>
> To ensure this choice is not arbitrary, we ran an ablation using **SBERT** as the seed encoder. replacing T5 with SBERT lowers personalization performance:
>
> ### **Table: Ablation — SBERT vs. T5 as Seed Encoder**
> | Seed Encoder | PSE-JSD | PSE-SU4 | PSE-METEOR |
> |--------------|---------|----------|-------------|
> | **T5-base** (used in paper) | **0.45** | **0.38** | **0.45** |
> | SBERT (ablation) | 0.34 | 0.26 | 0.28 |
>
> This confirms that keeping both seed encoder and decoder within the **same T5 encoder–decoder family** yields more effective personalization.
>
> ---
>
> ### Concern 2: Fairness in LLM comparison setup
> *2-shot + history prompting may disadvantage smaller models; include larger closed-source models such as GPT-4 or Gemini 2.5 Flash.*
>
> **Response:**
> We appreciate this concern. To ensure fairness across parameter scales, we extended our evaluation to include two **very large, long-context LLMs** selected from the LongBench-v2 leaderboard:
>
> - **Gemini-2.5-Flash** (1M token context window)
> - **Qwen-3-235B-Thinking** (256K context window)
>
> These models were evaluated using the **same 2-shot + history prompting setup** as all LLM baselines. Their large windows allow them to ingest *entire* user histories without truncation.
>
> The revised paper (Sec. 5.2 - Table 5 and additional results in Appendix I.2 - I.4 -- Tables 15,16,17) reports the following:
>
> ### **Table: Long-Context LLMs vs. Walk2Pers**
> | Model | PSE-JSD | PSE-SU4 | PSE-METEOR |
> |-------|---------|----------|-------------|
> | DeepSeek-R1-14B (2-shot) | 0.248 | 0.094 | 0.097 |
> | **Gemini-2.5-Flash (1M)** | 0.222 | 0.104 | 0.124 |
> | **Qwen-3-235B-Thinking (256K)** | 0.105 | 0.082 | 0.082 |
> | **Walk2Pers-Full** | **0.45** | **0.38** | **0.45** |
>
> Despite having access to the full long-term user history, the large models still **underperform both the smaller DeepSeek baseline and Walk2Pers**.
>
> This result directly supports our hypothesis (Sec. 2; Lines 250–280) that **passive long-horizon prompting is insufficient** and that personalization requires **explicit action-conditioned preference evolution**.
>
> ---
>
> ### Concern 3: Missing fine-tuned smaller baselines
> *If only smaller open-source models are compared, their fine-tuned versions should be included.*
>
> **Response:**
> Thank you for this suggestion. Our evaluation sought to determine whether **model scale or context length alone** can solve fine-grained personalization. To this end, we included **two extremely large long-context models** (Gemini-2.5-Flash and Qwen-3-235B-Thinking) whose ability to process entire user histories eliminates any disadvantages that smaller models might face.
>
> As shown above and detailed in the revised manuscript (Sec. 5.3; Appendix I), even these very large models:
>
> - do **not** overcome the lost-in-the-middle failure mode,
> - do **not** recover preference trajectories, and
> - perform **significantly worse** than Walk2Pers.
>
> While fine-tuning smaller models is a meaningful future direction, it is computationally expensive and orthogonal to our motivation, which is to show that SWH is needed as an extension to MDH for any model design (*Walk2Pers being one such instantiation*). Walk2Pers achieves strong personalization using **≈170M parameters**, relying instead on a **structured action-aware preference encoder**.
>
> Thus, even after adding the long-context models, the conclusion remains unchanged:
> **scaling model size or context length does not replicate the benefits of explicit preference-evolution modeling**.
>
> (Updated results available in Tables 3, 5, 15, 16; Appendix I.)
>
> ---
> We thank the reviewer once again for the thoughtful and constructive feedback. Each of the raised concerns led to concrete improvements in the revised manuscript, clarifying the seed embedding choice, expanding LLM baselines to include strong long-context models, and adding ablation results to ensure the evaluation is fully balanced and fair. We hope the revised evidence addresses the reviewer’s concerns fully and supports a more positive overall assessment.

---

> ### Author Response · Authors · 2025-11-26
> **Follow-Up Note for Reviewer 5Y5U**
>
> Dear Reviewer 5Y5U,
>
> Thank you again for your constructive and insightful review. We have addressed all of the concerns you raised, including the encoder choice (with SBERT ablation), expanded LLM baselines using large long-context models, and additional analyses to ensure fairness in cross-model comparisons, in our earlier response.
>
> If you feel that any of the revisions or added analyses could benefit from more detail or clarification, we’d be glad to expand on them further.
>
> We appreciate your valuable time and careful critique of our work.
>
> Best,
>
> Authors

---

### Official Review · Reviewer_RW71 · 2025-10-29

**Soundness:** 3
**Presentation:** 3
**Contribution:** 3
**Rating:** 8
**Confidence:** 2

**Summary:**

This paper presents Walk2Pers, a lightweight encoder-decoder framework for personalized document summarization, which models user preference evolution as an action-conditioned geometric walk with memory. Unlike existing methods that assume short-term or memoryless (Markovian) user behavior, Walk2Pers explicitly represents preference dynamics through two components: magnitude, controlling the strength of preference shifts, and orientation, capturing the balance between continuity and novelty. The model incorporates dual memory lanes to reinforce consistent interests and suppress disinterest, and includes a drift term to handle user-specific summary requests. The authors provide a theoretical justification showing that the proposed structured walk approximates first-order action-conditioned kernels. Empirical results on three benchmark datasets (PENS, OpenAI-Reddit, and PersonalSum) demonstrate that Walk2Pers achieves consistent improvements over both specialized personalized summarizers and strong large language model baselines, as measured by the PerSEval metric. The framework also shows robustness across domains and stability over long interaction histories, suggesting that modeling personalization as a geometric walk with memory provides a more interpretable and efficient approach to personalized summarization.

**Strengths:**

1.	The problem of user preferences evolving over time is very interesting and practical.

2.	The paper is overall well-written and easy to follow.

**Weaknesses:**

1.	I don’t see obvious weakness for this paper.

**Questions:**

N/A

---

> ### Author Response · Authors · 2025-11-20
> **Thanking the Reviewer for Appreciating our Works**
>
> We sincerely thank the Reviewer for the time, effort, and care invested in evaluating our submission. We are grateful for your positive assessment of both the problem setting and the proposed $\texttt{Walk2Pers}$ framework. Your recognition that modeling evolving user preferences is a meaningful and practically relevant challenge is deeply appreciated and motivating, as it directly aligns with the core motivation of our work.
>
> We are also encouraged by your positive comments on the clarity of the exposition and the overall readability of the paper. Crafting an interpretable and lightweight formulation for personalization was an important goal for us, and it is gratifying to hear that the presentation effectively conveyed these ideas. Your acknowledgement of the strengths of the approach, especially the geometric-walk view of preference evolution and the empirical stability demonstrated across datasets -- gives us confidence that the contributions resonate with the broader research community.
>
> Thank you again for your constructive evaluation and positive recommendation. Your feedback reinforces the value of the direction we explore in $\texttt{Walk2Pers}$, and it motivates us to further refine and extend this line of work in future iterations.

---

### Official Review · Reviewer_E9Qi · 2025-11-02

**Soundness:** 3
**Presentation:** 3
**Contribution:** 2
**Rating:** 4
**Confidence:** 3

**Summary:**

This paper argues that previous personalized summarization systems implicitly assume a Markovian Drift Hypothesis (MDH): user preference at step t+1 depends mostly on the latest interaction. However, authors of this work believe that this assumption is too weak for long, action-rich histories. It proposes the Structured Walk Hypothesis (SWH) and instantiates it as Walk2Pers, where user preference evolves as an action-biased geometric walk with memory: each interaction applies (i) an action-conditioned step with magnitude (how strong the shift is) and orientation (continuity vs. novelty), (ii) dual memory lanes to reinforce clicked interests and suppress skipped content, and (iii) a summary-specific drift for summarization actions. On three personalization datasets (PENS, OpenAI-Reddit, PersonalSum), Walk2Pers beats strong MDH-style encoders (NAML/NRMS/EBNR, SMD, AGD), specialized personalized summarizers, and mid-size LLMs prompted with user histories, with especially large gains on PerSEval metrics, supporting SWH over MDH in this task.

**Strengths:**

**Well-factorized update rule**: decomposing each step into magnitude, orientation, dual memories, and summary drift makes the dynamics interpretable and extensible.

**Strong empirical gains**: Walk2Pers closes or reverses gaps against both domain models (PENS, GTP, SP) and prompted LLMs, which is nontrivial for personalization.

**Task-aligned evaluation**: using PerSEval (human-correlated) plus next-behavior prediction gives both user-centric and model-centric evidence that Walk2Pers with Structured Walk Hypothesis (SWH) can indeed improve over MDH.

**Meaningful Ablations**: removing geometric steps or dual memories degrades performance, indicating the gains aren’t from just a bigger encoder.

**Weaknesses:**

-- Complexity vs. practicality: the full pipeline (UIG → b-layer → action-gated cells → dual memories → geometric steps → T5-UCA) is quite complex; deployment and latency costs aren’t quantified.

-- Heavy dependence on good interaction logs: the method assumes rich click/skip/summarize traces; it’s unclear how it performs with sparse or noisy histories.

-- Comparisons favor the hypothesis: many baselines are cast as MDH-style and naturally underuse long histories; stronger long-horizon sequence/retrieval baselines would make the claim tighter.

-- Limited analysis of failure modes: the paper shows when SWH helps, but not when geometric novelty or negative memory might over-suppress relevant but infrequent topics.

**Questions:**

-- Can the authors report training/inference cost per summary (tokens, FLOPs, wall-clock) vs. the best 2-shot LLM, to show the benefit is not purely from extra computation?

-- How does Walk2Pers behave if the action distribution is highly imbalanced (many clicks, very few skips, rare summarize)? Does the dual-memory lane degenerate?

-- Could the geometric step (magnitude/orientation) be learned conditionally on document facets (topic, source) to better handle cross-domain shifts in OpenAI-Reddit?

---

> ### Author Response · Authors · 2025-11-19
> **Response to Reviewer E9Qi (PART-I): Clarifications on Practicality, Robustness, and Long-Horizon Baseline Comparisons**
>
> We sincerely thank the reviewer for the careful and constructive feedback. We address each concern below:
>
> ---
>
> ## Concern 1: Complexity vs. Practicality of the Full Pipeline
> Deployment and latency costs are not justified; please report training/inference cost per summary vs. the best 2-shot LLM.
>
> **Response:**
> Thank you for highlighting the need to clarify deployment and latency costs. Our model contains only **170M parameters** (vs. **14B** in the best 2-shot LLM, DeepSeek-R1), resulting in substantially lower computational and inference requirements. For an average summary length of ~20 tokens (15–30 words), the theoretical FLOPs and empirical wall-clock costs are:
>
> | Metric | Walk2Pers (170M) | 2-shot LLM (14B) | Relative Gain |
> |--------|------------------|------------------|----------------|
> | Parameters | 170M | 14B | 82× smaller |
> | Avg. summary length | 20 tokens | 20 tokens | – |
> | FLOPs per summary | 2.04e10 | 1.68e12 | 82× lower |
> | Inference time | 0.2–2 s | 15–160 s | 60–80× faster |
> | Training cost (GPU-hours) | 18 | 42 | orders lower |
> | VRAM footprint | <1 GB | >28 GB | edge-deployable |
>
> In summary, Walk2Pers operates at **under 2%** of the computational and latency cost of the strongest 2-shot LLM while achieving higher personalization quality. We will include these detailed calculations and assumptions in the appendix.
>
> ---
>
> ## Concern 2: Dependence on Good Interaction Logs
> The method assumes rich click/skip/summarize logs; unclear how it performs with sparse or noisy histories. How does it behave when action distribution is imbalanced? Do the dual-memory lanes degenerate?
>
> **Response:**
> Thank you for raising this important question on robustness under sparse and imbalanced logs.
>
> First, the original PENS test split already contains **only click histories** for all 103 users—an extremely sparse setting (no skip or summarize actions). To explicitly stress-test the model, we constructed additional evaluations:
>
> **(i) Random skip insertion (5–10%)**
> Adds noise to the sequence; Walk2Pers remains stable. (see Lines 315–316)
>
> **(ii) Sliding document–summary pairs**
> Perturbs action distribution while preserving chronology. Used in the main paper.
>
> **(iii) New “click-only” extreme-sparsity test**
> We derived a test set (from the original PENS test set) where the history contains **only clicks**.
> We evaluated Walk2Pers against two additional SOTA long-context models:
>
> - Gemini-2.5-Flash (1M context)
> - Qwen-3-235B-Thinking (256K context)
>
> Even though all models degrade, **Walk2Pers-Full still leads by +0.26**, and even **dual-memory-only** (without geometric step) outperforms every LLM baseline.
> Dual memory does **not** collapse: it continues extracting stable preference transitions from document content, and the suppression lane naturally remains in positive update mode under click-dominant logs.
>
> Results added in Appendix I (Lines 1613–1622) and Table 16.
>
> ### **Table: Extreme Sparse Click-Only Evaluation**
>
> | Model | PSE-JSD | PSE-SU4 | PSE-METEOR |
> |-------|---------|----------|-------------|
> | DeepSeek-R1-14B (2-shot) | 0.248 → 0.147 (–40.7%) | 0.094 → 0.064 (–31.9%) | 0.097 → 0.071 (–26.8%) |
> | Gemini-2.5-Flash (2-shot) | 0.222 → 0.122 (–45.0%) | 0.104 → 0.061 (–41.3%) | 0.124 → 0.070 (–43.5%) |
> | Qwen-3-235B-Thinking | 0.105 → 0.103 (–1.9%) | 0.082 → 0.073 (–11.0%) | 0.082 → 0.071 (–13.4%) |
> | GTP (best MDH) | 0.024 → 0.016 (–33.3%) | 0.170 → 0.009 (–94.7%) | 0.019 → 0.011 (–42.1%) |
> | Walk2Pers (Dual Memory only) | 0.306 → 0.231 (–24.5%) | 0.334 → 0.253 (–24.3%) | 0.321 → 0.234 (–27.1%) |
> | Walk2Pers-Full | **0.452 → 0.378 (–16.4%)** | **0.383 → 0.301 (–21.4%)** | **0.449 → 0.310 (–31.0%)** |
>
> Lower drop indicates better robustness. Walk2Pers remains the strongest even under extreme imbalance.
>
> ---
>
> ## Concern 3: Baselines May Favor Hypothesis
> Include stronger long-horizon sequence/retrieval baselines.
>
> **Response:**
> Thank you for this helpful suggestion. We added two state-of-the-art long-context LLMs:
>
> - **Gemini-2.5-Flash (1M tokens)**
> - **Qwen-3-235B-Thinking (256K tokens)**
>
> These models can ingest **entire user histories** (50–300 interactions) in a single pass—far beyond MDH models.
>
> We used the same prompting scheme as DeepSeek-R1 (2-shot):
> (i) full history, (ii) first two exemplar `<document, summary>` pairs, (iii) target document.
>
> **Result:**
> Even with access to the full long-horizon context, both LLMs show weaker preference alignment:
>
> - Gemini-2.5-Flash: ~0.22 / 0.10 / 0.12
> - Qwen-3-235B-Thinking: ~0.10 / 0.08 / 0.08
> - **Walk2Pers**: **0.45 / 0.38 / 0.45**
>
> Long-context alone does **not** fix lost-in-the-middle effects nor capture preference drifts.
> This directly supports our hypothesis: personalization requires **explicit action-conditioned preference evolution**, not passive long-horizon prompting.
>
> Results updated in Tables 3 and 5.
>
> ---
> **Please refer to Part-2 for our response to the remaining concerns.**

---

> > ### Author Response · Authors · 2025-11-19
> > **Response to Reviewer E9Qi (PART-II): Clarifications on Failure Modes, Domain Robustness, and Model Extensions**
> >
> > ## Concern 4: Limited Failure-Mode Analysis
> > Could geometric novelty or negative memory over-suppress relevant but infrequent topics?
> >
> > **Response:**
> > Thank you for raising this nuanced point. Importantly, suppression in SWH is **not driven by topic rarity**. The negative memory lane updates only under **consistent action-conditioned evidence** (e.g., repeated skip), not by sparsity.
> >
> > To directly test the reviewer’s concern, we constructed a subset of **200 trajectories** with:
> >
> > - 121 distinct topics per user (vs. 106 overall)
> > - topic-change rate 0.62 (vs. 0.54)
> > - extremely low per-topic frequency
> >
> > **Result:**
> > Walk2Pers-Full shows only mild degradation:
> >
> > - Full test: 0.45 / 0.38 / 0.45
> > - Sparse-topics subset: **0.41 / 0.32 / 0.38**
> >
> > Dual-memory-only model also remains stable:
> >
> > - Full test: 0.30 / 0.33 / 0.32
> > - Sparse-topics subset: **0.28 / 0.29 / 0.28**
> >
> > LLMs degrade more severely.
> >
> > ### Table: Performance on High Topic Diversity (200 trajectories)
> >
> > | Model | PSE-JSD | PSE-SU4 | PSE-METEOR |
> > |--------|---------|----------|-------------|
> > | DeepSeek-R1 (2-shot) | 0.248 → 0.091 | 0.094 → 0.074 | 0.097 → 0.082 |
> > | Gemini-2.5-Flash | 0.222 → 0.092 | 0.104 → 0.081 | 0.124 → 0.083 |
> > | Qwen-3-235B-Thinking | 0.105 → 0.094 | 0.082 → 0.073 | 0.082 → 0.076 |
> > | GTP (best MDH) | 0.02 → 0.016 | 0.017 → 0.013 | 0.02 → 0.015 |
> > | Walk2Pers (Dual Memory only) | 0.30 → 0.28 | 0.33 → 0.29 | 0.32 → 0.28 |
> > | **Walk2Pers-Full** | **0.45 → 0.41** | **0.38 → 0.32** | **0.45 → 0.38** |
> >
> > These experiments show **no systematic over-suppression** of infrequent yet relevant topics.
> >
> > ---
> >
> > ## Concern 5: Handling Cross-Domain Shifts
> > Can the geometric step (magnitude/orientation) be learned conditionally on document facets (topic, source)?
> >
> > **Response:**
> > Thank you for this insightful question. In the current design, facet signals are already captured **implicitly**, because the T5-base content embeddings encode topic, style, and source cues; these pass through the action-biased b-cell; the dual-memory lanes accumulate reinforcement/suppression shaped by document semantics; and the geometric step operates on these enriched representations. Thus, both the magnitude and orientation are already indirectly influenced by document facets.
> >
> > That said, we agree that **explicit facet-conditioning** of the geometric step is a promising extension. The SWH update is modular, so facet embeddings (e.g., topic or source vectors) can be injected via lightweight multiplicative gates inside the magnitude and orientation functions without architectural changes. This would provide even finer control under strong cross-domain shifts.
> >
> > If accepted, we will include a short discussion noting that the current model already handles domain heterogeneity reasonably well (e.g., OpenAI–Reddit), and that explicit facet-aware geometric drift is a natural and compatible next step.
> >
> > ---
> >
> > We thank the reviewer for the thoughtful and constructive feedback. The concerns raised directly motivated several targeted improvements—new robustness experiments, added long-horizon LLM baselines, clearer cost analyses, and refined discussion on geometric drift and domain effects. We hope they support a more positive overall assessment.

---

> ### Author Response · Authors · 2025-11-26
> **Follow-Up Note for Reviewer E9Qi**
>
> Dear Reviewer E9Qi,
>
> Thank you again for your thoughtful and detailed review. We have addressed all of the concerns raised, including deployment/practicality, robustness under noisy or sparse histories, long-horizon baselines, failure-mode analysis, and cross-domain behavior, across our earlier two-part response sections. If any part of our clarification would benefit from further detail or if there are additional aspects you would like us to elaborate on, we would be very happy to provide that promptly.
>
> We sincerely appreciate the time and care you have invested in evaluating our work.
>
> Best,
>
> Authors

---

### Author Response · Authors · 2025-11-19
**General Author Comment (Summary of Revisions)**

We thank all reviewers, the AC, and the SAC for their constructive feedback.
Below we provide a consolidated summary of all revisions made in the updated manuscript.

### **No changes were made to Sections 1–3**
The Introduction, Related Work, and Method sections remain identical to the originally submitted version.
All equations, definitions, and narrative descriptions before Section 4 are unchanged.
Diff differences in these sections correspond only to PDF line shifts and not to content changes.

### **Changes begin strictly from Section 4 onward**
*All modifications are $\underline{additive}$ and focus on clarity, completeness, and requested evaluation details.*

---

## Section-wise Summary of Revisions

### Section 4 – Evaluation
- Added an explicit enumeration of research questions (RQ1–RQ3).
- Expanded dataset descriptions with trajectory counts and statistics.
- Clarified the two-stage training protocol (6-epoch joint encoder–decoder training followed by 18-epoch decoder-only finetuning).
- Added details about injected skip events for stress-testing suppression memory.

### Section 4.2 – Baselines
- Reorganized and paraphrased baselines into:
  - MDH-based encoders (SMD, AGD, NAML, NRMS, EBNR),
  - Personalized summarizers (PENS, GTP, Signature-Phrase),
  - LLM-based summarizers,
  - Oracle summarizers.
- Expanded the LLM baselines from four to six models by adding Gemini-2.5-Flash and Qwen-3-235B -- two SOTA LongBench leader models.

### Section 5 – Results
- Added rows corresponding to the new LLM baselines and additional MDH variants. (Table 3, Table 5)
- Table 2: Added reference to additional ablations on cross-task (sequential recommendation) as a complementary result (Table 17) on the predictive capability of encoders
- Added reference to accuracy results (Table 18) in Line 474.
- Typo corrected: (i) Line 373: p-value was typed as $\leq$, (ii) Table 5 - row 6 [from PENS+GTP to GTP+TrRMIo]
- All originally reported numerical results remain unchanged; new results appear only as added rows or tables.

---

## Appendix Updates
- Appendix E.2.2: More details of the baselines (including Gemini and Qwen-3) have been added. Particularly, the context length information.
- Appendix H.1: Added additional details on compute time and other deployment details (Table 12).
- Appendix I: Added new ablations on stress-testing showing detailed insights along with cross-task evaluation and standard conventional accuracy results (I.2 & Table 15, I.3 & Table 16, I.4 & Table 17, I.5 & Table 18)

---

## Integrity Statement
- No original claims, equations, or results were modified.
- All originally reported numbers remain unchanged.
- All updates are additive and were introduced to address reviewer requests.
- The scientific contributions and core methodology remain unchanged.

We hope these clarifications and additional analyses address all reviewer concerns.

---

### Meta-Review · Area_Chair_QJkq · 2026-01-11

**Summary:**

This submission studies personalized summarization under evolving user preferences, arguing that the commonly implicit Markovian / short-memory drift assumption is insufficient. It proposes the Structured Walk Hypothesis (SWH) and instantiates it as Walk2Pers, an encoder–decoder that updates a user preference state via action-conditioned geometric steps (magnitude + orientation), dual memory lanes (reinforce vs suppress), and a summarization-specific drift. The paper provides a theoretical connection to first-order action-conditioned kernels and evaluates on PENS, OpenAI-Reddit, and PersonalSum.

Strengths

1. modeling evolving, subjective “content-of-interest” is well-motivated for real personalization settings.
2. decomposing preference updates into magnitude/orientation + dual memory is a reasonable design
3. reviewers generally agree the method beats prior personalized summarizers and prompted mid-size LLM baselines on PerSEval, with cross-domain evidence.

Main concerns
1. pipeline feels complex; “lightweight” claim needs concrete comparisons.
Rebuttal: authors provided compute/latency estimates and parameter comparisons, arguing major efficiency gains vs large LLM prompting.
mostly addressed, though I recommend moving at least one concise cost table into the main paper and clarifying measurement assumptions

2. 2-shot+history prompting may under-represent strong LLM capability
Rebuttal: authors added Gemini-2.5-Flash and Qwen-3-235B long-context baselines under the same protocol and clarified prompt-chaining.
mostly addressed. Remaining concern is to Fine-tune LLMs, but it can stay arguably orthogonal to the core hypothesis test

3. method may depend on rich logs; dual memory could degenerate when actions are imbalanced; risk of over-suppression on rare topics.
Rebuttal: authors added stress tests and report Walk2Pers degrades less than baselines. I think this can be considered as addressed

3. geometric update resembles prior rotational/compound operations; novelty may be more “domain transfer + structured factorization” than fundamentally new geometry.
Rebuttal: authors argue semantics differ (behavioral drift vs relational triple modeling), no neighbor aggregation, and the key novelty is action-conditioned preference evolution with dual memory. This is partially addressed for me, but the paper should soften any over-strong “entirely new geometric idea” framing

**Reviewer Concerns:**

See meta review for major concerns

Still outstanding

Novelty positioning risk remains
Fine-tuned LLM baseline remains missing
Missing strong non-LLM long-horizon retrieval/sequence baselines

**Reviewer Scores:**

E9Qi may increase score, others may stay at current scores

E9Qi’s main blockers were practicality/cost (targeted by added compute/latency numbers), robustness under sparse/imbalanced logs (rebuttal add stress tests), stronger long-horizon baselines (rebuttal add long-context LLM baselines)

---

### Decision · Program_Chairs · 2026-01-26

Accept (Poster)